# On the Importance of Uncertainty in Decision-Making with Large Language Models

**Nicolò Felicioni**[*]                                          *nicolo.felicioni@polimi.it*
*Politecnico di Milano*

**Lucas Maystre**                                               *lucasm@spotify.com*
*Spotify*

**Sina Ghiassian**                                              *sinag@spotify.com*
*Spotify*

**Kamil Ciosek**                                                *kamilc@spotify.com*
*Spotify*

**Reviewed on OpenReview:** *https://openreview.net/forum?id=YfPzUX6DdO*

## Abstract

We investigate the role of uncertainty in decision-making problems with natural language as input. For such tasks, using Large Language Models as agents has become the norm. However, none of the recent approaches employ any additional phase for estimating the uncertainty the agent has about the world during the decision-making task. We focus on a fundamental decision-making framework with natural language as input, which is the one of contextual bandits, where the context information consists of text. As a representative of the approaches with no uncertainty estimation, we consider an LLM agent with a greedy policy, which picks the action corresponding to the largest predicted reward. We compare this baseline to LLM agents that make active use of uncertainty estimation by integrating the uncertainty in a Thompson Sampling policy. We employ different techniques for uncertainty estimation, such as Laplace Approximation, Dropout, and Epinets. We empirically show on real-world data that the greedy policy performs worse than the Thompson Sampling policies. These findings suggest that, while overlooked in the LLM literature, uncertainty improves performance on bandit tasks with LLM agents.

## 1 Introduction

Large language models (LLMs) have emerged as a dominant paradigm in natural language processing (Ouyang et al., 2022; OpenAI, 2023), achieving state-of-the-art performance across a wide range of tasks (Rae et al., 2021). To reach this progress, LLMs have pushed model scale and dataset size to unprecedented levels. By optimizing such immense models exclusively to predict text, perhaps surprisingly, LLMs have achieved strong performance on a broad range of datasets and tasks (Bubeck et al., 2023), including translation, question-answering, and dialogue.

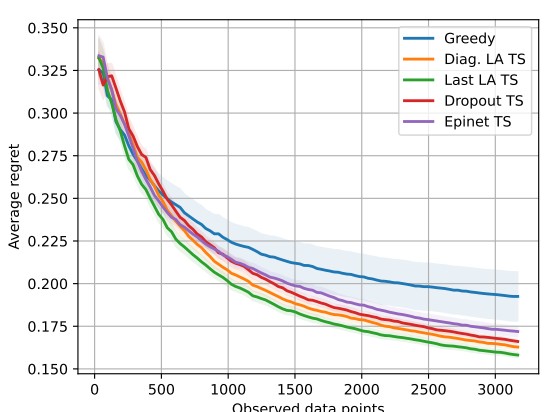

Figure 1: Average regret obtained on toxic content detection bandit task.

---

[*]Research completed while on internship at Spotify.

In parallel, many critical real-world systems are increasingly relying on these models to make decisions (Yang et al., 2023b), where the consequences of a particular *action* are typically reflected by a *reward signal*. This usually works by fine-tuning an LLM to serve as a reward model, where we try to teach the network to predict the mean reward for each action. A common way of using this model to make decisions is to use the predictions greedily (Riquelme et al., 2018), taking the action with the greatest estimated reward. However, this approach ignores the fact that reward estimates may be inaccurate, which can lead to never-ending pathological behavior (Russo et al., 2018). On the other hand, the literature on *contextual bandits* (Wang et al., 2005) provides a principled approach to deal with inaccurate reward estimates in a decision-making task. A popular modelling tool for doing this in the Bayesian framework is to maintain a probabilistic reward model that relies on two separate notions of uncertainty: *epistemic* and *aleatoric*.

Epistemic uncertainty is the one that reflects the fact that we have not yet seen enough fine-tuning data to estimate the mean reward well, while aleatoric uncertainty reflects the irreducible noise associated with observing a reward. When such a reward model is coupled with an action-selection algorithm like *Thompson Sampling* (Thompson, 1933), high-quality estimates of epistemic uncertainty can be leveraged to produce a better balance between exploration and exploitation in the decision-making task, evidenced by lower regret.

In this paper, we want to investigate the importance of employing epistemic uncertainty models in decision-making tasks with natural language and LLMs. In particular, we focus on one of the most fundamental decision-making tasks with natural language as input: *contextual bandits*. In a contextual bandit problem, the agent must continually take actions based on observed context, which in our case is text, and observes the reward only for the chosen action. This framework is applicable to a myriad of real-world scenarios.

As a concrete example application, let us consider the problem of automated content moderation on online platforms (Gorwa et al., 2020; Vaccaro et al., 2020; Ma & Kou, 2021; Avadhanula et al., 2022). In this scenario, users post comments and the agent decides whether to publish each comment. The user comments serve as the context, in text form. The agent takes an action (e.g. "publish" or "not publish") based on the context. After acting, the environment provides the reward only for the selected action, which the agent uses to update its policy. For example, if the agent publishes a toxic comment, it receives a large negative reward since other users will report the comment. Instead, if the agent does not publish a comment, it receives a neutral reward regardless of the comment's content, since the comment is not seen. To learn effectively in this setting, the agent must balance exploration and exploitation.

In this paper, we study the contextual bandit scenario empirically, combining the insights from the LLM and bandit communities. The greedy approach outlined above is our main baseline and was chosen for its simplicity and ease of implementation. We benchmark it against approximate Bayesian models coupled with the Thompson Sampling algorithm for action selection. Since Thompson Sampling requires epistemic uncertainty to be of high quality to work well, we investigate several different approaches of doing so. Specifically, we compare techniques for epistemic uncertainty estimation that can be scaled up to be used with current LLMs. These are the Laplace Approximation (Daxberger et al., 2021), Dropout (Gal & Ghahramani, 2016) and Epinets (Osband et al., 2023a). We show that all our TS policies significantly and consistently outperform the greedy baseline on a real-world bandit benchmark in terms of final regret. A preview of the results is shown in Figure 1. These findings suggest that, while often overlooked in the LLM literature, epistemic uncertainty improves performance on bandit tasks with LLM agents.

**Contributions**

- We provide an extensive empirical study of LLM-based contextual bandits;

- We adapt existing neural epistemic uncertainty estimation techniques to LLMs. In particular, we identify the best variant of the Laplace approximation and dropout to use, and we show one possible epinet architecture that can also tackle this task;

- We empirically show on real-world data that the greedy policy is sub-optimal in terms of final regret, and that epistemic uncertainty is helpful in contextual bandit problems with text data.

To the best of our knowledge, this is the first time that such an empirical analysis on bandit learning with LLMs has been provided. This empirical evaluation sheds light on critical aspects of LLMs, such as their performance on decision-making tasks and the importance of epistemic uncertainty for those models.

## 2 Preliminaries

In this section, we describe the problem statement and we introduce the topic of uncertainty in Machine Learning models.

### 2.1 Batch Contextual Bandit problem

The contextual bandit problem is a sequential decision-making framework where an agent interacts with an environment over a number of time steps. In particular, we focus on the more general case where, for each time step, we observe a batch of contexts and the agent has to select a batch of actions (Kandasamy et al., 2018).

Let us call the total number of time steps $T$. At each time step $t = 1, 2, \ldots, T$, the agent observes a batch of contexts, which we denote with $x_t^1, \ldots, x_t^B$. Each context $x_t^b$ contains useful information that should be used to solve the task. In this paper, we focus on decision-making problems driven by text. Hence, in this case, each context $x_t^b$ consists of text information. The agent then uses the observed data until now, which we call $\mathcal{D}$, to select an action $a_t^b$ from a set $\mathcal{A}$ of $K$ possible actions, conditioned on the given context $x_t^b$ with $b \in \{1, \ldots, B\}$. For each selected action $a_t^b$, the agent receives a reward $r_t^b$, that depends on the context $x_t^b$ and action $a_t^b$. At the end of the time step, we have observed a set of tuples that we denote as $\mathcal{D}_t = \{(x_t^1, a_t^1, r_t^1), \ldots, (x_t^B, a_t^B, r_t^B)\}$. This set of tuples is added to the observed dataset $\mathcal{D} \leftarrow \mathcal{D} \cup \mathcal{D}_t$. The aim of the agent is to learn a policy that minimizes the *regret* over the $T$ time steps. We define the average regret as the difference between the cumulative reward from the learned policy and the cumulative reward from an *optimal* policy, all divided by the number of time steps. We define the optimal policy $\pi^*$ as the policy that, within the considered policy space $\Pi$, maximizes the expected reward: $\pi^* = \arg\max_{\pi \in \Pi} \mathbb{E}_x \mathbb{E}_{a \sim \pi(\cdot|x)} [r(x, a)]$. Let us call $r_t^*$ the expected rewards obtained by the optimal policy at any time step $t$. We can define the average regret as follows:

$$R_T = \frac{1}{T} \sum_{t=1}^{T} \left( r_t^* - \frac{1}{B} \sum_{b=1}^{B} r_t^b \right) .$$

Since we focus on bandits with text information as contexts, we would like to use the state-of-the-art natural language processing capabilities of pre-trained LLMs and build bandit policies based on these models. We will explain in more detail how to design such deep bandits in Section 3.

### 2.2 Aleatoric and Epistemic Uncertainty in Machine Learning

In the field of machine learning, the concept of uncertainty plays a crucial role in both model interpretation and reliability. A common framework to characterize uncertainty, given a particular model class, is to categorize it into two types: *aleatoric* and *epistemic*. Aleatoric uncertainty arises from the inherent randomness in the data or the environment. It represents the variability in the outcome that cannot be reduced even if more data is provided. Hence, this type of uncertainty is irreducible and intrinsic to the current task or dataset. On the other hand, epistemic uncertainty stems from the lack of knowledge. This type of uncertainty is reducible by collecting more data. It reflects the uncertainty in the model parameters due to limited data or limited model capacity.

As a concrete example, consider a contextual bandit setting with observed data $\mathcal{D} = \{(x_1, a_1, r_1), \ldots\}$ and a parametric model $f_\theta(x, a)$ that we wish to train to predict expected reward given context and action, with parameters $\theta$. In this regression setting, a common approach to model the aleatoric uncertainty is to assume that there is additive noise in the rewards: $r = \mathbb{E}[r] + \epsilon$, where the noise is distributed according to a generic distribution $\epsilon \sim P(\epsilon)$. A simple but effective choice is to set the noise distribution to be Gaussian: $\epsilon \sim \mathcal{N}(0, \sigma_{\text{obs}}^2)$, where $\sigma_{\text{obs}}^2$ is the *observation variance*. This kind of uncertainty is independent of the model, so it is irreducible even with infinite data.

---

**Algorithm 1** Greedy

---
**Require:** Bandit model $f_\theta$.
1: Initialize $\mathcal{D} \leftarrow \emptyset$
2: **for** time $t = 1, \ldots, T$ **do**
3:     Observe contexts $x_t^1, \ldots, x_t^B$.
4:     **for** $b = 1, \ldots, B$ **do**
5:         Select $a_t^b = \arg\max_a f_\theta(x_t^b, a)$.
6:         Observe reward $r_t^b$.
7:     **end for**
8:     Create $\mathcal{D}_t = \{(x_t^1, a_t^1, r_t^1), \ldots, (x_t^B, a_t^B, r_t^B)\}$
9:     Add to the observed dataset $\mathcal{D} \leftarrow \{\mathcal{D}_1, \ldots, \mathcal{D}_t\}$.
10:    Update the parameters $\theta$ training on the observed data $\mathcal{D}$ minimizing Eq. 3.
11: **end for**

---

Epistemic uncertainty, instead, stems from uncertainty in the model parameters $\theta$. A principled Bayesian approach is to maintain a posterior distribution over the parameters, $P(\theta|\mathcal{D})$, that is updated as new data arrives. The posterior covariance quantifies epistemic uncertainty. As the dataset grows, the posterior distribution becomes more peaked, reducing the epistemic uncertainty.

Maintaining a posterior over $\theta$ allows the model to "know what it does not know", critical for balancing exploration and exploitation in bandits. We will expand on using Bayesian techniques for principled exploration in Section 3.

## 3 Large Language Model Bandits

In this section, we describe how to build bandit agents with pre-trained Large Language Models.

We approach the contextual bandit problem with a regression model. Specifically, we train a model to predict the expected reward for each action, given the context. To leverage recent advances in large pre-trained language models, we initialize our model with a pre-trained LLM. As these models are pre-trained for next-token prediction, which is a classification task, we discard the final classification layer and append a new linear regression output layer to predict expected rewards.

Specifically, let $\pi_{\theta_{\mathrm{PT}}}$ denote the pre-trained LLM with parameters $\theta_{\mathrm{PT}}$. Such a model is a function that maps any sequence of tokens into a probability distribution over the vocabulary: $\pi_{\theta_{\mathrm{PT}}}(x) \in \Delta(V)$, where $\Delta(V)$ denotes the simplex over the vocabulary of tokens $V$. For our task, we do not need the final classification head; hence, we consider only the features before the final layer, which we denote as $\tilde{\pi}_{\theta_{\mathrm{PT}}}(x) \in \mathbb{R}^d$. Now, for each tuple $(x, a, r)$ in our dataset $\mathcal{D}$, we feed $x$ into the pre-trained feature extractor $\tilde{\pi}$ and we pass those features through a final linear layer to produce the expected reward for each $a \in \mathcal{A}$. We call our model $f_\theta$, which is defined as follows:

$$f_\theta(x) = \mathrm{linear}(\tilde{\pi}_{\theta_{\mathrm{PT}}}(x)) \in \mathbb{R}^K \ , \tag{1}$$

where $K = |\mathcal{A}|$. Also, we denote the $a$-th output of our model as $f_\theta(x, a) \in \mathbb{R}$.

### 3.1 Greedy policy

As a representative approach with no additional uncertainty estimation phase, we consider the greedy policy. The behavior of this policy is illustrated in Algorithm 1. In particular, in our case, we use a pre-trained LLM as a feature extractor with a linear layer on top to create a regression model $f_\theta$, as described in Section 3. Given a context, the greedy policy selects the action for which the model predicts the highest reward. Let us consider a certain time step $t$. After the action selection phase, we observe the real reward for the batch of actions the policy has selected. The dataset will be composed of $t$ sets of tuples: $\mathcal{D} = \{\mathcal{D}_1, \ldots, \mathcal{D}_t\}$. We can exploit the observed data to update the parameters of the regression model before the beginning of the

next time step. A typical way to do so is to compute a maximum-a-posteriori (MAP) estimate. This is done by minimizing a loss corresponding to the negative log posterior:

$$\mathcal{L}(\theta; \mathcal{D}) = \underbrace{\sum_{(x,a,r) \in \mathcal{D}} l(\theta; x, a, r)}_{\text{negative log-likelihood}} + \underbrace{r(\theta)}_{\text{negative log-prior}} . \tag{2}$$

Usually, for regression task, it is assumed that there is Gaussian aleatoric noise with zero mean and fixed variance $\sigma_{\text{obs}}^2$. This implies that the negative log-likelihood part of the loss is a mean-squared error (MSE): $l(\theta; x, a, r) = \frac{1}{2\sigma_{\text{obs}}^2}(r - f_\theta(x, a))^2$. Regarding the prior, another typical assumption is to set the prior belief on the parameters as indepentent Gaussians with 0 mean and $\sigma_p^2$ variance. However, in our case, it is inconsistent to assume that all the weights have 0 prior since we initialize every weight (except for the last layer) to have the same weight as a pre-trained LLM. Hence, we use a different prior mean: $\theta_p = [\theta_{\text{PT}}, 0]$, where with the notation $[v_1, v_2]$ we denote the concatenation of vectors $v_1$ and $v_2$. Therefore, this means that every weight (except for the last layer) has a prior centered in the pre-trained weight value, while the last layer is regularized towards zero. The negative log-prior term hence becomes[1] $r(\theta) = \frac{1}{2\sigma_p^2}||\theta - \theta_p||_2^2$.

Minimizing the loss shown in Equation 2 at the end of each time step, however, can have some drawbacks in a bandit task. Indeed, it can be highly expensive for large neural models, especially because the complete dataset is constantly increasing at each time step. An alternative and more scalable loss to minimize is the following, which considers only the data $\mathcal{D}_t$ observed at the current time step $t$:

$$\mathcal{L}(\theta^{(t)}; \mathcal{D}_t) = \sum_{b=1}^{B} (r_t^b - f_{\theta^{(t)}}(x_t^b, a_t^b))^2 + \lambda ||\theta^{(t)} - \theta^{(t-1)}||_2^2 , \tag{3}$$

where $\lambda = \sigma_{\text{obs}}^2 / \sigma_p^2$, and we set $\theta^{(0)} = \theta_p$. This loss now only assumes that the data points in $\mathcal{D}_t$ are i.i.d., and we update the prior at each time step with the weights obtained at the previous time step. This will be the loss we will use to optimize the greedy bandit in our experimental analysis in Section 5. Using this loss, we are doing MAP inference many times, applying the Bayes rule anew for each new batch of data. In this way, we perform MAP, which is a well-known established principle in deep learning, while avoiding looping throughout the whole dataset. For a more detailed discussion, we refer to Appendix B. This loss is also present in the literature. For instance, (Daxberger et al., 2021) use this same loss in their continual learning experiment (Daxberger et al. (2021), Appendix C.4.1).

## 3.2 Thompson Sampling

*Thompson Sampling* (TS) (Thompson, 1933; Russo et al., 2018) is a probabilistic algorithm for the contextual bandit problem. The key idea of Thompson Sampling is to maintain an epistemic uncertainty estimate in the form of a posterior distribution over the parameters of the model $P(\theta|\mathcal{D})$. Initially, this distribution is set to a prior distribution $P(\theta)$ that represents the agent's initial uncertainty. On each time step, for each observed context, the agent samples a set of parameters $\hat{\theta} \sim P(\theta|\mathcal{D})$ from the posterior distribution, and selects the action with the highest reward according to the sampled model. At the end of the time step, the agent updates the posterior distribution of the parameters with the observed data. A high-level description of this procedure is provided in Algorithm 2.

This approach balances exploration and exploitation by sampling from a posterior that quantifies the epistemic uncertainty. Actions with more uncertain estimates will be explored, while actions that are currently believed to have higher rewards will be exploited. TS can be thought as taking actions according to the (epistemic) probability that they are optimal, which leads to a good balance between exploration and exploitation. The posterior distribution concentrates over time, automatically adjusting the exploration-exploitation trade-off.

Thompson Sampling has been shown to be effective both in theory and in practice (Chapelle & Li, 2011). When the Bayesian updates are exact, there are theoretical guarantees on the performance of TS agents

---

[1]This technique was also proposed in (Xuhong et al., 2018), and it has been shown to reduce the fine-tuning loss when using pre-trained neural models.

---

**Algorithm 2** Thompson Sampling

---

**Require:** Bandit model $f_\theta$.
**Require:** Prior distribution on the parameters $P(\theta) \leftarrow \mathcal{N}(\theta_p, \Sigma_p)$.
 1: Initialize $\mathcal{D} \leftarrow \emptyset$
 2: **for** time $t = 1, \ldots, T$ **do**
 3:     Observe context $x_t^1, \ldots, x_t^B$.
 4:     **for** $b = 1, \ldots, B$ **do**
 5:         Sample parameters $\hat{\theta} \sim P(\theta|\mathcal{D})$.
 6:         Select $a_t^b = \arg\max_a f_{\hat{\theta}}(x_t^b, a)$.
 7:         Observe reward $r_t^b$.
 8:     **end for**
 9:     Create $\mathcal{D}_t = \{(x_t^1, a_t^1, r_t^1), \ldots, (x_t^B, a_t^B, r_t^B)\}$
10:     Add to the observed dataset $\mathcal{D} \leftarrow \mathcal{D} \cup \mathcal{D}_t$.
11:     Update the posterior distribution $P(\theta|\mathcal{D})$.
12: **end for**

---

(Agrawal & Goyal, 2017). For instance, Russo & Van Roy (2016) show that Thompson Sampling has a Bayesian regret bound. However, in our case, we are dealing with a complex bandit model that uses a pre-trained LLM. Therefore, an exact Bayesian update is computationally infeasible, and we have to resort to approximations. In Section 4, we will show different techniques to estimate posterior distribution of the parameters of a neural network and how to adapt such techniques to our case of LLM bandits.

## 4 Epistemic uncertainty estimation for pre-trained LLMs

In order to use a Thompson Sampling policy, we need to estimate the epistemic uncertainty of our model. This additional step is not required for a greedy policy. Also, it is not a trivial step for deep neural networks, for which there is no (tractable) closed-form solution to update the posterior distribution of the parameters.

Crucially, for our particular task, the uncertainty estimation technique is required to scale to very large models, which are typically pre-trained at great expense, such as LLMs. This means that we cannot employ techniques that require us to change the training process (Graves, 2011), or ensembles (Lakshminarayanan et al., 2017), which require us to run training multiple times. Instead, we rely on more scalable techniques, such as Dropout, Laplace Approximation, and Epinets. In the following, we discuss these epistemic uncertainty estimation techniques in further detail, and we show how to adapt them to LLM bandits.

### 4.1 Dropout

The *dropout* technique (Srivastava et al., 2014) consists of randomly setting a proportion $p$ of neuron outputs to zero at each forward pass through the network during training. This random "dropping out" of neurons allows the network to sample and train on different (but overlapping) architectures.

In standard supervised learning, dropout is then deactivated during inference, and all dropout neurons are re-scaled to account for the fact that all dropout neurons are active. However, in our case, we still apply dropout during the action selection (Gal & Ghahramani, 2016; Riquelme et al., 2018). Using dropout in this phase, we randomly select (according to the dropout probability) a set of parameters $\hat{\theta}$ to use. This procedure can be seen as sampling parameters from an approximate posterior distribution: $\hat{\theta} \sim P(\theta|\mathcal{D})$, and it has connections with variational inference (Gal & Ghahramani, 2016). With such approximate posterior distribution, we can apply Thompson Sampling. This technique is valuable for large models, such as our LLM bandit, because it does not add any overhead to the procedure and does not require additional memory.

### 4.2 Laplace Approximation

The *Laplace Approximation* (LA) is a technique that can be used to approximate the posterior distribution of the parameters of a neural model and does not require changing the training process or training multiple models.

LA exploits the fact that, from a Bayesian point of view, minimizing a regularized loss can be seen as finding a *maximum-a-posteriori* (MAP) estimate: $\theta_{\mathrm{MAP}} = \arg\min_\theta \mathcal{L}(\mathcal{D}; \theta)$.

Then, LA consists of replacing the loss with its second-order Taylor expansion around $\theta_{\mathrm{MAP}}$:

$$\mathcal{L}(\mathcal{D}; \theta) \approx \mathcal{L}(\mathcal{D}; \theta_{\mathrm{MAP}}) + \frac{1}{2}(\theta - \theta_{\mathrm{MAP}})^T H (\theta - \theta_{\mathrm{MAP}}), \quad H = \nabla^2 \mathcal{L}(\mathcal{D}; \theta)|_{\theta_{\mathrm{MAP}}} \tag{4}$$

Notice that the first-order term vanishes because we are expanding around $\theta_{\mathrm{MAP}}$, which is a point of minimum of the loss. After some algebraic manipulations, it can be shown that the posterior distribution can be expressed as[2]:

$$P(\theta|\mathcal{D}) = \mathcal{N}(\theta_{\mathrm{MAP}}, H^{-1}) , \tag{5}$$

which means that, after training, the posterior distribution of the parameters is a Gaussian distribution, centered on the parameters obtained with training (i.e., $\theta_{\mathrm{MAP}}$), and with the inverse Hessian as covariance matrix. Therefore, to derive the approximate posterior in practice, we need first to identify the weights $\theta_{\mathrm{MAP}}$ by training our LLM bandit regression model. Subsequently, the only additional step is the calculation of the Hessian matrix $H$. This posterior distribution can be used to apply the Thompson Sampling policy by sampling a set of parameters $\hat{\theta} \sim P(\theta|\mathcal{D})$ at each time step. Notice, however, that there are still significant drawbacks in using LA with LLM bandits: (1) computing the Hessian requires looping through the whole fine-tuning dataset, which is always increasing; (2) storing the Hessian is infeasible due to its size, which is quadratic with the number of parameters; (3) computing the Hessian may be computationally infeasible, and the Hessian may be indefinite. In the following, we show that there are many ways to cope with these limitations.

**Recursive computation of the Hessian** By applying Bayesian reasoning, we notice that the Hessian at a given time step $t$ can be computed recursively, exploiting the Hessian computed at $t-1$. Let us assume that we are at time step $t$. Hence, the dataset will look like this: $\mathcal{D} = \{\mathcal{D}_1, \ldots, \mathcal{D}_t\}$, where each $\mathcal{D}_i$ is composed of i.i.d. data points. The posterior distribution of the parameters can be re-written as follows:

$$P(\theta|\mathcal{D}) \propto P(\mathcal{D}_t|\theta) \cdot \ldots \cdot P(\mathcal{D}_2|\theta) \cdot \underbrace{\underbrace{P(\mathcal{D}_1|\theta) \cdot P(\theta)}_{\propto P(\theta|\mathcal{D}_1)}}_{\propto P(\theta|\mathcal{D}_1, \mathcal{D}_2)}, \tag{6}$$

where $P(\mathcal{D}_t|\theta) = \prod_{b=1}^{B} P(r_t^b | x_t^b, a_t^b, \theta)$. This implies that:

$$P(\theta|\mathcal{D}_1, \ldots, \mathcal{D}_t) \propto P(\mathcal{D}_t|\theta) P(\theta|\mathcal{D}_1, \ldots, \mathcal{D}_{t-1}). \tag{7}$$

Hence, the Hessian of the negative log-posterior at time $t$, which we call $H^{(1:t)}$, can be re-written as:

$$H^{(1:t)} = \underbrace{\nabla^2 - \log P(\mathcal{D}_t \mid \theta)|_{\theta_{\mathrm{MAP}}^{(t)}}}_{\text{neg. log-likelihood Hessian } H_l^{(t)}} + \underbrace{\sum_{t'=1}^{t-1} \nabla^2 - \log P(\mathcal{D}_{t'} \mid \theta)|_{\theta_{\mathrm{MAP}}^{(t')}}}_{\text{previous neg. log-likelihood Hessians } H_l^{(t')}} + \underbrace{\nabla^2 - \log P(\theta)|_{\theta_{\mathrm{MAP}}^{(t)}}}_{\text{neg. log-prior Hessian } H_p} . \tag{8}$$

This means that, at the end of any time step $t$, we just need to compute the log likelihood Hessian $H_l^{(t)}$ with respect to the current data $\mathcal{D}_t$ and sum the previous Hessian: $H^{(1:t)} = H_l^{(t)} + H^{(1:t-1)}$.

---

[2]For a more detailed derivation, see Appendix C.

Furthermore, this formulation gives rise to a loss that uses only the current data when training at time $t$:

$$\mathcal{L}(\theta^{(t)}; \mathcal{D}_t) = \underbrace{\frac{1}{2\sigma_{\text{obs}}^2} \sum_{(x,a,r)\in\mathcal{D}_t} \left(r - f_\theta^{(t)}(x,a)\right)^2}_{\text{neg. log likelihood: } -\log P(\mathcal{D}_t|\theta)} + \underbrace{\frac{1}{2}(\theta^{(t)} - \theta_{\text{MAP}}^{(t-1)})^T H^{(1:t-1)}(\theta^{(t)} - \theta_{\text{MAP}}^{(t-1)})}_{\text{neg. log prior / updated posterior: } -\log P(\theta|\mathcal{D}_1,\ldots,\mathcal{D}_{t-1})} \quad . \quad (9)$$

If we are at time $t = 1$, we assume that $\theta_{\text{MAP}}^{(0)}$ are the initial weights, and $H^{(1:0)} = \nabla^2 \log P(\theta)|_{\theta_{\text{MAP}}^{(t)}}$.

**Diagonal approximation**   Even if computed recursively, storing the full Hessian may be infeasible even for small neural networks. Hence, it is not a viable option for the case of LLM bandits. A practical approximation is to consider only the *diagonal* of the matrix, which is equivalent to maintain a posterior distribution for each parameter independently. This reduces the memory complexity from quadratic to linear in the number of parameters.

**Fisher Hessian approximation**   To further reduce the computational complexity, we can replace the true Hessian by the *expected Fisher*[3] matrix. If we assume a model where the aleatoric noise is Gaussian with mean zero and fixed variance $\sigma_{\text{obs}}^2$, we can compute this approximation of the Hessian as follows:

$$\text{diag}(\hat{H}_l^{(t)}) = \frac{1}{\sigma_{\text{obs}}^2} \sum_{(x,a)\in\mathcal{D}_t} (\nabla f_\theta(x,a))^2 \quad (10)$$

This approximation has the advantage of being positive semi-definite and (Kunstner et al., 2019) showed that it is accurate when the regression residuals are small. For more details on this approximation, see Appendix C.

### 4.3   Last-Layer Laplace Approximation

Alternatively, instead of using Hessian approximations, another way to scale LA for large models is to compute the full Hessian but only for a subset of the parameters of the network, which typically is the last layer of the model. In our case, this means that we will compute the Hessian for the randomly initialized final layer, while the pre-trained LLM layers remain fixed during the sampling phase of TS (every parameter is still fine-tuned during training). This approximation can reduce the exploration due to the fixed weights, but at the same time it allows a full quadratic Hessian computation, leading to a better quality of exploration. Last layer LA is equivalent to *Bayesian linear regression* (Box & Tiao, 2011) on the features before the last layer within a single batch. Also, with Gaussian likelihood, last-layer LA is exact within a single batch.

### 4.4   Epinets

A different approach to estimating epistemic uncertainty is the *epinet* (Osband et al., 2023a). An epinet is a heuristic approach that tries to estimate the epistemic uncertainty with a separate neural network. It consists of a neural network added to a base network, which, in our case, is the LLM bandit model. The epinet takes as inputs both features $\tilde{\pi}_\theta(x)$ derived from the base network and an *epistemic index $z$*, which is a random vector sampled from a reference distribution $P_Z$. In the case of our LLM bandit model, $\tilde{\pi}_\theta(x)$ denotes the feature vector extracted by the model before the final regression layer. The prediction is then obtained by adding the predictions of the base network and the epinet:

$$g_{\theta,\eta}(x; z) = f_\theta(x) + \text{epi}_\eta(\text{sg}[\tilde{\pi}_\theta(x)], z) \ ,$$

where with sg we denote the stop-gradient operator, and with $\eta$ we denote the additional parameters of the epinet.

---

[3]We use the term expected Fisher to stress the fact that it is different from the empirical Fisher matrix (Kunstner et al., 2019).

The epinet $\text{epi}_\eta$ comprises two parts: a learnable network $\text{epi}_\eta^L$ and a prior network $\text{epi}^P$ that represents prior uncertainty. The prior network has no trainable parameters. This allows the epinet to adapt uncertainty estimates to observed data. In principle, every network that takes features and epistemic index as inputs could be an epinet. However, for our use case, we need to use a small epinet so that we do not add excessive computational overhead to the LLM bandit model. In Section 5, we describe in further detail the epinet architecture we selected for our experiments.

During training, the epinet model $g_{\theta,\eta}$ is trained as a normal deep learning model, with the addition of a sampling phase of an epistemic index $z \sim P_Z$ for each data point. Also during the action selection phase, for each action to select, an epistemic index $z$ is sampled. While this is not a Bayesian approach, the sampling phase can be seen as an approximate Thompson Sampling, as shown by Osband et al. (2023b).

## 5 Experiments

In this section, we provide an extensive empirical study of LLM-based contextual bandits on real-world data. We show how to adapt the epistemic uncertainty estimation techniques we described to bandits with pre-trained LLMs. In particular, we identify the best variant of the Laplace approximation and dropout to use, in addition to a proposed Epinet architecture that we show works well for TS with LLM bandits. Finally, we empirically show that the greedy policy is sub-optimal (in terms of final regret), shedding light on the importance of epistemic uncertainty for LLM bandits.

### 5.1 Experimental Methodology

**Tasks**  We evaluate the bandit policies on various real-world tasks. These tasks consist of:

- An open-source dataset, called "*measuring hate speech*", which is openly available on *HuggingFace Datasets*[4]. This dataset consists of around 136,000 comments. Each of them is associated with a continuous score, called the "hate speech score", provided for each comment. Comments with a score $> 0.5$ are considered "toxic" (around 36% of the comments are labeled as toxic, while the others are not toxic). We will refer to this task as `toxic`.

- An open-source dataset, called "*IMDb*" (Maas et al., 2011), which is openly available on *HuggingFace Datasets*[5]. This dataset consists of 50,000 movie reviews. Each of them is associated with a *class*, which can be "positive" or "negative", according to the sentiment expressed in the review. The dataset consists of exactly 50% "positive" reviews and 50% "negative" reviews. We will refer to this task as `imdb`.

- An open-source dataset, called "*Offensive Language Identification*" (Zampieri et al., 2019), which is openly available on *HuggingFace Datasets*[6]. This dataset consists of over 14,000 English tweets. Each of them is classified as "non-offensive" or "offensive". The dataset consists of 66.9% "non-offensive" tweets and 33.1% "offensive" tweets. We will refer to this task as `offensive`.

- An open-source dataset, called "*HatEval*" (Basile et al., 2019), which is openly available on *HuggingFace Datasets*[7]. This dataset consists of 13,000 English and Spanish tweets, where some of them contain hate speech against immigrants and women. Each of them is classified as "non-hateful" or "hateful". The dataset consists of 58% "non-hateful" tweets and 42% "hateful" tweets. We will refer to this task as `hate`.

These tasks are framed as a contextual bandit problem as follows. The context is the text input, and the actions available to the agent are "not publish" or "publish". For each time step, the agent will observe a batch of $B = 32$ comments. The reward function is the following: if the agent decides not to publish a

---

[4]`https://huggingface.co/datasets/ucberkeley-dlab/measuring-hate-speech`

[5]`https://huggingface.co/datasets/imdb`

[6]`https://huggingface.co/datasets/tweet_eval/viewer/offensive`

[7]`https://huggingface.co/datasets/tweet_eval/viewer/hate`

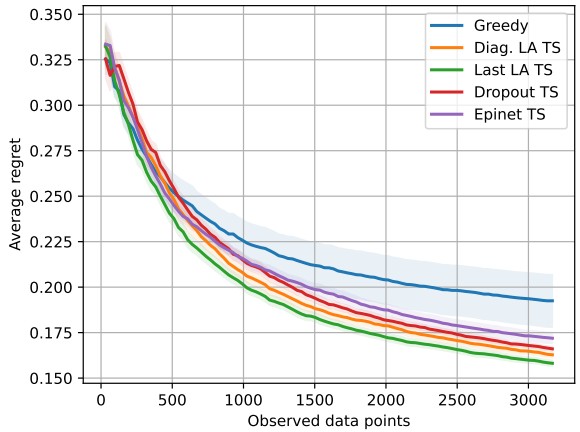

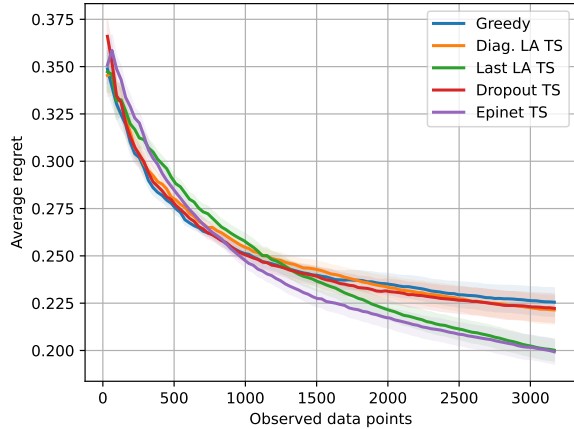

Figure 2: Average regret ($\pm$ std. err.) obtained on the `toxic` bandit task.

Figure 3: Average regret ($\pm$ std. err.) obtained on the `imdb` bandit task.

comment, a reward of 0.5 is observed regardless of the actual toxicity of the comment. If the agent publishes a non-toxic comment, a reward of 1 is observed. If the agent publishes a toxic comment, a reward of -0.5 is observed. This asymmetric reward function represents a possible example suitable for this real-world scenario. The goal of the agent is to minimize the regret (compared to a clairvoyant optimal publishing policy) over time by learning to make optimal publish/not publish decisions based on the text content. For the `imdb` task, we consider "negative" reviews as toxic content; for the `offensive` task, we consider "offensive" tweets as toxic content; for the `hate` task, we consider "hateful" tweets as toxic content.

**Bandit models** To investigate the role of uncertainty in LLM bandits, we compare different TS variants with the greedy baseline. Every bandit model is initialized as in Eq. 1, with a pre-trained GPT2 model with 124M parameters (Radford et al., 2019). We also include additional experiments with GPT2-XL (1.5B) in order to investigate if our findings generalize to larger models.

As TS variants, we include dropout, Diagonal Fisher LA (which we will call Diag. LA), LA with full Hessian on the last layer (which we will call Last LA), and Epinet TS. Regarding the epinet architecture, we follow prior work on epinets (Osband et al., 2023a;b) and select an architecture with a multi-layer perceptron $h$ which is multiplied with a dot product with the epistemic index: $\text{epi}_\eta(x; z) = h_\eta([\text{sg}(\tilde{\pi}_\theta(x)), z])^T z$. For more details on the architecture, see Appendix D. Every model is trained with regularized MSE loss as in Eq. 3, except for Diag. LA, which updates the prior with the new posterior at each time step, as shown in Eq. 9. All the parameters are updated during training. We train each model at the end of each time step for 50 epochs with the *Adam* optimizer (Kingma & Ba, 2014), with learning rate set to $3 \cdot 10^{-5}$. For each model, hyperparameters are tuned on 10 random runs (with different seeds than the testing ones) with $T = 50$ on the `toxic` task. We did not tune the dropout probability because we wanted to exploit the fact that GPT2 was pre-trained with dropout $p = 0.1$ and use the same $p$. We investigate the effectiveness of this choice in the results section. We describe the training phase and the hyperparameter tuning procedure in more detail in Appendix D.

### 5.2 Results

We show the experimental results (20 random runs, $T = 100$) for the `toxic` task (Figure 2), `imdb` task (Figure 3), `offensive` task (Figure 4), and `hate` task (Figure 5) with GPT2 (124M) LLM bandits. These results empirically exhibit the importance of actively using epistemic uncertainty in contextual bandit problems with Large Language Models. Overall, we notice how the approaches leveraging epistemic uncertainty tend to achieve lower regret compared to the greedy policy in all the considered tasks.

Analyzing the results in more detail, we observe that Dropout TS exhibits strong performance on the `toxic` and `hate` tasks, achieving lower regret compared to the greedy policy. However, its performance is comparable to the greedy approach on the `imdb` and `offensive` tasks. This behavior is expected, as Dropout TS and the

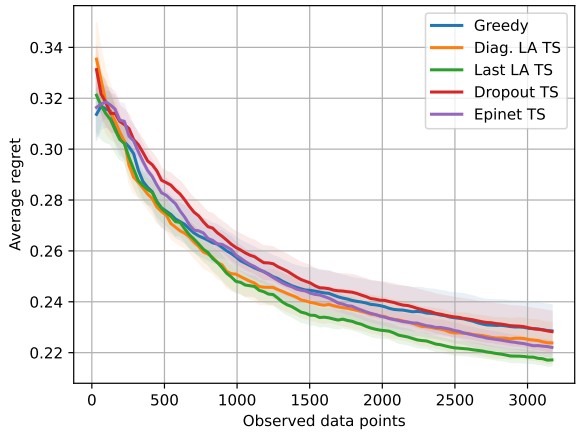

Figure 4: Average regret (± std. err.) obtained on the `offensive` bandit task.

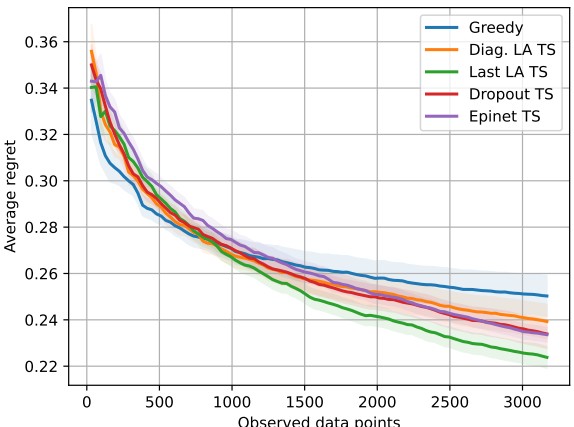

Figure 5: Average regret (± std. err.) obtained on the `hate` bandit task.

greedy policy are very similar, with the only distinction being the activation of dropout during inference in Dropout TS. It is important to notice that we did not tune the dropout probability, using the same one used during the pre-training of the base LLM. The performance of Dropout TS may possibly improve after tuning the dropout probability. In Appendix D, we investigated the effect of changing the dropout probability. Also, Dropout TS incurs no additional computational cost, requiring only a single forward pass, akin to the greedy method.

Diag. LA TS demonstrates excellent results on the `toxic` task and attains lower regret than the greedy policy on the `offensive` and `hate` tasks. However, its performance on the `imdb` task is comparable to the greedy approach. This may be attributed to the fact that hyperparameter selection was performed on the `toxic` task, and it may not generalize well to the `imdb` task.

Both Last LA TS and Epinet TS consistently achieve lower regret compared to the greedy policy across all four tasks, albeit with some distinctions. Last LA TS emerges as the best-performing method in all cases, but it requires computing the full Hessian on the last layer, which can be computationally expensive. On the other hand, Epinet TS, while slightly inferior to Last LA TS, offers a more computationally efficient alternative by employing an additional small neural network. These results confirm previous findings in the literature (Osband et al., 2022; 2023b). Also, we conjecture that the choice of the epinet architecture may influence results. Indeed, epinets are a very broad class of neural networks, and exploring epinet design was not the focus of this work. Hence, we did not perform an extensive architecture search, but we believe that our work can shed light on the potential of epinets, and we will leave further investigation to future work.

Moreover, we notice that the confidence interval of the greedy policy is generally large. This suggests that there is a high variance in the results obtained by the greedy policy. To further investigate the cause of this finding, let us define the action selection ratio as $s_T(a) = \frac{1}{T} \sum_{t=1}^{T} \mathbb{I}(a_t = a)$. We selected two sample runs for the `toxic` task and plotted the action selection ratio for the action "publish" during these two sample runs in Figure 6. Notice that the action "publish" is a risky action, meaning that it can lead to a strongly negative reward if toxic content is published, while the action "not publish" is safer and gives a constant reward. Figure 6 clearly shows that there are cases where the greedy policy suffers from under-exploration and persists in choosing a suboptimal arm, while Thompson Sampling (in this case, Diag. LA) exhibits a more balanced behavior. Due to this issue, the greedy algorithm will incur constant regret in the random runs which require more balanced exploration. This issue makes the greedy policy perform worse than TS policies in our experiments.

**Experiments with 1.5B LLM** To further investigate if our findings generalize to larger models, we performed additional experiments using GPT2-XL, a larger language model with 1.5B parameters. For this set of experiments, we train our models for 5 epochs for each batch of data. We perform this new set of experiments on the `hate` task. The results of these experiments (20 random runs, $T = 100$) are shown

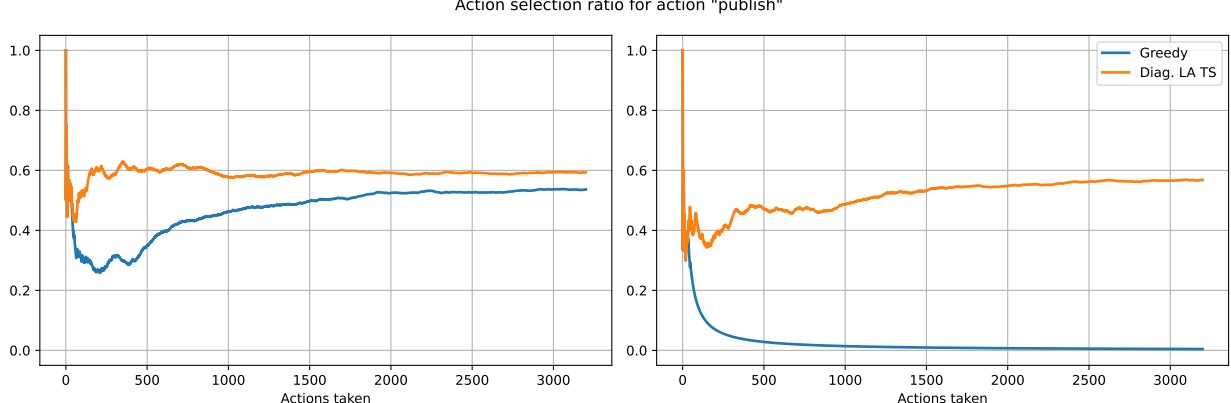

Figure 6: Action selection ratio for the action "publish" for two particular sample runs (`toxic` task).

in Figure 7. These results confirm our findings, showing how TS policies generally achieve a lower regret compared to the greedy policy. In particular, we again see how Last LA TS achieved the lowest regret among the TS variants. These results further highlight the importance of leveraging epistemic uncertainty in contextual bandit problems, even when scaling up to larger language models.

**Additional ablation study on model dimension and pre-training** We also performed an additional ablation study to investigate the effect of using large pre-trained models to initialize the bandit policies in our experiments. We conduct those studies by running a bandit algorithm using a smaller transformer model with weights initialized from scratch. We report those experiments in Appendix D.4.

## 6 Conclusion

In this paper, we investigated the role of epistemic uncertainty estimation in decision-making tasks that use natural language as input. For such tasks, using Large Language Models as agents has become the norm. However, none of the recent ap-

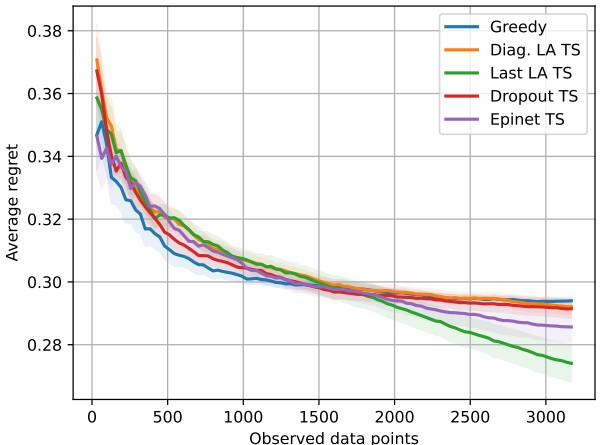

Figure 7: Average regret ($\pm$ std. err.) obtained on the `hate` bandit task with 1.5B LLM bandits.

proaches estimates the epistemic uncertainty of the agent. We focused on a fundamental decision-making task: the contextual bandit problem, where context consists of text. We approached the bandit task with a deep regression model initialized with a pre-trained LLM. As a representative of the approaches with no uncertainty estimation, we considered an LLM bandit with a greedy policy, which picks the action corresponding to the largest predicted reward. We compared the greedy baseline with various approaches integrating uncertainty estimates into the decision process via Thompson Sampling. We adapted several epistemic uncertainty estimation techniques to LLMs, such as dropout, Laplace Approximation, and epinets. Finally, we provided an empirical analysis of bandit learning with LLMs on real-world data. Our experiments showed that using uncertainty information leads to greatly improved performance over the greedy approach. These improvements highlight the benefits of modeling uncertainty for exploration in bandit problems with text and Large Language Models. Our work suggests that uncertainty should play a more central role in developing LLM-based agents for decision-making. As future work, a natural extension would be to use our study on

LLM bandits as a stepping stone to more complex settings like text-based reinforcement learning. Another interesting direction would be to study the scaling behavior of LLM bandits in a systematic way.

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

## A    Prior Work

**Decision-making with Large Language Models**   Large language models (LLMs) recently emerged as a dominant paradigm in natural language processing (Ouyang et al., 2022; OpenAI, 2023), achieving state-of-the-art performance across a wide range of tasks (Rae et al., 2021), pushing model scale and dataset size to unprecedented levels. Models such as the OpenAI's GPT LLM series (Radford et al., 2018; 2019; Brown et al., 2020; OpenAI, 2023), Google's PaLM (Chowdhery et al., 2023) and Gemini (Gemini Team et al., 2023), or Meta's LLaMA (Touvron et al., 2023a;b) have leveraged the transformer architecture (Vaswani et al., 2017) with model sizes ranging from hundreds of millions to hundreds of billions of parameters and are trained on up to hundreds of billions of text examples.

Due to their remarkable capabilities in text processing, LLMs have also been applied to decision-making tasks (Yang et al., 2023b), and there is a plethora of papers in the research literature that investigated this idea (Li et al. 2022; Carta et al. 2023; Chen et al. 2023; Klissarov et al. 2023; *inter alia*). One of the most prominent examples is that of dialogue agents. Many recent papers model the dialogue between the LLM and a user as a sequential decision-making problem, where the action is the answer that the LLM should provide to the user after receiving the user's message. In particular, those works typically use *Reinforcement Learning* (RL) techniques (Ouyang et al., 2022) to fine-tune language models for dialogue applications. Hence, they use the LLM as a policy to solve the RL problem with state-of-the-art RL algorithms, such as *Proximal Policy Optimization* (Schulman et al., 2017). Another example is enhancing LLMs by allowing them to use external *tools* (Thoppilan et al., 2022; Yang et al., 2023a; Hao et al., 2023; Gao et al., 2023; Mialon et al., 2023; Schick et al., 2023). In this case, the LLM-based agent has an action space which is the set of external tools at disposal and the various interactions possible with those tools.

The aforementioned approaches typically assume that a sufficient amount of data has been collected a priori to train an effective policy. They do not explicitly address the exploration-exploitation trade-off and they do not systematically explore issues around the role of epistemic uncertainty in decision-making. In contrast, in our work, we explicitly consider the problem of learning from interaction, and we focus on one of the most fundamental and natural decision-making tasks, which is the one of *contextual bandits* (Chapelle & Li, 2011). We comprehensively investigate the role of epistemic uncertainty for bandit models with pre-trained LLMs. The importance of exploiting epistemic uncertainty is emphasized also in (Dwaracherla et al., 2024), where they propose an approach to approximate epistemic uncertainty for the alignment problem of LLMs.

**Uncertainty in Deep Learning**   Deep learning models provide state-of-the-art performance in several different tasks, ranging from image recognition to natural language processing. However, these models usually provide poor uncertainty estimates (Kendall & Gal, 2017). For this reason, several techniques have been proposed in the literature that allow the estimation of the uncertainty of deep learning algorithms. A useful mathematical model for characterizing uncertainty in deep learning is to categorize it into two types: epistemic and aleatoric (Gal, 2016; Hüllermeier & Waegeman, 2021). Epistemic uncertainty stems from our lack of knowledge about the best model to describe a process. It is reducible as more data or knowledge is gathered. Aleatoric uncertainty, in contrast, is due to the inherent randomness in the data or environment and is irreducible even with more data. In particular, epistemic uncertainty has been proven to be particularly helpful when it comes to decision-making problems.

A possible way to capture epistemic uncertainty in deep learning is to equip neural networks with a distribution over the parameters, which is then updated upon seeing new data. This kind of neural network is usually called *Bayesian Neural Network* (BNN). From a theoretical perspective for decision making under uncertainty, Bayesian neural networks (BNNs) are appealing since they provide a full posterior distribution over models, allowing the derivation of formal regret bounds to guide exploration (Agrawal & Goyal, 2017). However, exact BNN inference is intractable for large models like LLMs. Thus, we must rely on approximations.

One common approach to approximate the posterior distribution is Dropout (also called Monte Carlo Dropout) (Gal & Ghahramani, 2016). Dropout is a technique initially proposed for standard supervised learning (Srivastava et al., 2014), which consists of randomly setting a proportion $p$ of neuron outputs to zero at each forward pass through the network during training. In standard supervised learning, dropout

is then deactivated during inference, and all dropout neurons are re-scaled to account for the fact that all dropout neurons are active. However, we can still apply dropout during inference. In this way, we randomly select (according to the dropout probability) a set of parameters $\hat{\theta}$ to use. This procedure can be seen approximately as sampling parameters from a posterior Bayesian distribution, and it is proven to have links with variational inference techniques (Gal & Ghahramani, 2015).

Another popular technique is Variational Inference (VI) (Graves, 2011). VI is a technique in machine learning used for approximating complex posterior distributions in Bayesian inference. The goal is to find a simpler, parameterized distribution (the variational distribution) that is close to the true posterior distribution of interest by solving an optimization problem. However, it requires changes in the training procedure, which is not possible if we want to use pre-trained models.

Laplace Approximation (LA) (MacKay, 1992) is another technique that is used to approximate the posterior distribution of the parameters of a neural network. It consists of assuming that the training consists of a maximum-a-posteriori estimate of the parameters and replacing the loss with its second-order Taylor expansion around the MAP estimate. With these two steps, an analytical solution for the posterior distribution of the weights can be found. In particular, the weights are distributed as a multivariate Gaussian with the MAP estimate as the mean and the inverse of the Hessian as the covariance matrix. The key advantage of LA is that the MAP estimate of the weights is usually available after standard deep learning training. A more problematic issue is the computation of the Hessian. However, a recent paper (Daxberger et al., 2021) investigates different techniques to approximate the Hessian, making the computation feasible even for modern neural networks.

There are also non-Bayesian approaches to estimating epistemic uncertainty. *Deep Ensembles* (Lakshminarayanan et al., 2017) provide a conceptually simple way to capture uncertainty, but are expensive and do not yield a well-defined posterior. The authors show that the degree of disagreement among the NNs within the ensemble is indicative of the epistemic uncertainty of the ensemble. While promising, this approach is not suitable for LLMs, for which an ensemble would be prohibitively expensive for both training and inference.

A more affordable approach is the use of epinets (Osband et al., 2023a). An epinet estimates the epistemic uncertainty with a separate neural network that takes as inputs both features derived from the base network (usually before the last layer) and an *epistemic index*, which is a random vector sampled from a fixed reference distribution. The prediction is then obtained by adding the predictions of the base network and the epinet. Though not Bayesian, epinets has been shown to provide useful uncertainty estimates for guiding exploration combined with a Thompson Sampling policy, with much lower overhead than ensembles (Osband et al., 2023b).

In our paper, we focus on scalable approaches to estimating epistemic uncertainty, and we adapt them to the case of pre-trained LLM: we used the pre-trained weights as prior, exploited the dropout probability used in the pre-training phase, and used different Hessian approximations. Finally, we empirically show the importance of using epistemic uncertainty by embedding it into Thompson Sampling policies (Thompson, 1933). This significantly outperforms a greedy policy that does not account for uncertainty in decision-making.

## B  Additional details on the used loss

In this section, we elaborate in more detail the loss described in Eq. 3.

Using the loss shown in Eq. 3, we are doing MAP inference many times, applying the Bayes rule anew for each new batch of data. In this way, we perform MAP, which is a well-known established principle in deep learning, while avoiding looping throughout the whole dataset. Indeed, at a given time step $t$, the dataset will look like this: $\mathcal{D} = \{\mathcal{D}_1, \ldots, \mathcal{D}_t\}$, where each $\mathcal{D}_i$ is composed of i.i.d. data points. The posterior distribution of the parameters can be re-written as follows:

$$P(\theta|\mathcal{D}) \propto P(\mathcal{D}_t|\theta) \cdot \ldots \cdot P(\mathcal{D}_2|\theta) \cdot \underbrace{\underbrace{P(\mathcal{D}_1|\theta) \cdot P(\theta)}_{\propto P(\theta|\mathcal{D}_1)}}_{\propto P(\theta|\mathcal{D}_1, \mathcal{D}_2)}, \tag{11}$$

where $P(\mathcal{D}_t|\theta) = \prod_{b=1}^{B} P(r_t^b|x_t^b, a_t^b, \theta)$. This implies that:

$$P(\theta|\mathcal{D}_1, \ldots, \mathcal{D}_t) \propto P(\mathcal{D}_t|\theta)P(\theta|\mathcal{D}_1, \ldots, \mathcal{D}_{t-1}). \tag{12}$$

This justifies the loss in Eq. 3 as a MAP method: it is equivalent to minimizing the negative logarithm of Eq. 12.

## C   Additional details on Laplace Approximation

In this section, we describe in more detail the Laplace Approximation technique.

**Laplace Approximation**   The Laplace Approximation (LA) technique can be used to approximate the posterior distribution of weights of a neural network. As we will see, Laplace Approximation does not require any change in the training process nor training multiple models; thus, it is feasible for modern deep learning neural networks (such as LLMs) that are typically pre-trained (hence, their training procedure can not be modified) and expensive to train (hence, training multiple models is unfeasible).

Typically, deep neural networks are trained via the minimization of a loss. Let us say that we are given a dataset $\mathcal{D} = \{(x_i, y_i)\}_{i=1}^{D}$. We call the neural network we want to train $f_\theta$, which is a parametric function. The parameters of the neural network are $\theta \in \mathbb{R}^N$. Standard deep learning losses can usually be decomposed in a regularization term and a sum of empirical loss terms on single data points:

$$\mathcal{L}(\mathcal{D}; \theta) = r(\theta) + \sum_{i=1}^{D} l(x_i, y_i; \theta) \tag{13}$$

From a Bayesian point of view, it is straightforward to interpret the regularization term as the negative log prior: $r(\theta) = -\log P(\theta)$, and the sum of empirical losses as the negative log-likelihoods: $\sum_{i=1}^{D} l(x_i, y_i; \theta) = -\log P(\mathcal{D}|\theta) = -\sum_{i=1}^{D} \log P(y_i|f_\theta(x_i))$. This means that minimizing such a loss is actually leading to a *maximum-a-posteriori* (MAP) estimate: $\theta_{\mathrm{MAP}} = \arg\min_\theta \mathcal{L}(\mathcal{D}; \theta)$. From these considerations, we can also re-write the posterior distribution as follows:

$$P(\theta|\mathcal{D}) = \frac{1}{Z}P(\mathcal{D}|\theta)P(\theta) = \frac{1}{Z}\exp(-\mathcal{L}(\mathcal{D}; \theta)), \quad Z = \int P(\mathcal{D}|\theta)P(\theta)d\theta \tag{14}$$

Now, Laplace Approximation consists in replacing the loss with its second-order Taylor expansion around $\theta_{\mathrm{MAP}}$:

$$\mathcal{L}(\mathcal{D}; \theta) \approx \mathcal{L}(\mathcal{D}; \theta_{\mathrm{MAP}}) + \frac{1}{2}(\theta - \theta_{\mathrm{MAP}})^T H(\theta - \theta_{\mathrm{MAP}}), \quad H = \nabla_\theta^2 \mathcal{L}(\mathcal{D}; \theta)|_{\theta_{\mathrm{MAP}}} \tag{15}$$

Notice that the first-order term vanishes because we are expanding around $\theta_{\mathrm{MAP}}$, which is a point of minimum of the loss.

Starting from this approximation, it can be shown that the posterior distribution is a multivariate Gaussian. First, we obtain a closed-form solution for the normalizing constant $Z$:

$$
\begin{aligned}
Z &= \int P(\mathcal{D}|\theta)P(\theta)d\theta = \int \exp(-\mathcal{L}(\mathcal{D}; \theta))d\theta \\
&\approx \int \exp(-\mathcal{L}(\mathcal{D}; \theta_{\mathrm{MAP}}) - \frac{1}{2}(\theta - \theta_{\mathrm{MAP}})^T H(\theta - \theta_{\mathrm{MAP}}))d\theta \\
&= \exp(-\mathcal{L}(\mathcal{D}; \theta_{\mathrm{MAP}})) \int \exp(-\frac{1}{2}(\theta - \theta_{\mathrm{MAP}})^T H(\theta - \theta_{\mathrm{MAP}}))d\theta \\
&= \exp(-\mathcal{L}(\mathcal{D}; \theta_{\mathrm{MAP}})) \frac{(2\pi)^{\frac{N}{2}}}{(\det H)^{\frac{1}{2}}} ,
\end{aligned}
$$

where the last equality derives from the multivariate normal density.

Now, if we come back to the posterior distribution, we obtain:

$$
\begin{aligned}
P(\theta|\mathcal{D}) &= \frac{1}{Z}P(\mathcal{D}|\theta)P(\theta) = \frac{1}{Z}\exp(-\mathcal{L}(\mathcal{D};\theta)) \\
&\approx \frac{(\det H)^{\frac{1}{2}}}{(2\pi)^{\frac{N}{2}}}\exp(-\frac{1}{2}(\theta-\theta_{\mathrm{MAP}})^T H(\theta-\theta_{\mathrm{MAP}})) \\
&= \mathcal{N}(\theta;\theta_{\mathrm{MAP}},H^{-1})
\end{aligned}
\tag{16}
$$

Therefore, to derive the approximate posterior in practice, we need first to identify the weights $\theta_{\mathrm{MAP}}$ that maximize the log-posterior function. This, in deep learning terms, corresponds to the training phase, where a regularized loss is minimized. Subsequently, the only additional step is the calculation of the Hessian matrix at the point $\theta_{\mathrm{MAP}}$. This means that LA can also be applied to pre-trained neural networks with no need to change the training procedure. This is a fundamental requirement if we want to apply this procedure to LLMs.

**Expected Fisher Matrix** The Laplace approximation as outlined above requires the knowledge of the Hessian of the loss function. Typically, a further approximation is used. First of all, let us rewrite the Hessian as the sum of the Hessian of the likelihood and the Hessian of the prior: $H = H_l + H_p$. Now, the true Hessian of the likelihood is replaced by the expected Fisher[8] matrix.

$$
H_l \approx |\mathcal{D}|\,\mathbb{E}_x\mathbb{E}_{y\sim P_\theta(y|x)}\left[-\nabla^2\log P_\theta(y|x)\right]
\tag{17}
$$

Despite the fact that the Hessian on the left hand side depends on the true regression targets, while the right hand side does not, the approximation is accurate when the regression residuals are small (Kunstner et al., 2019). In order to make the formula in equation 17 practical, we replace the expectation with respect to the datapoints using a Monte-Carlo estimate.

$$
|\mathcal{D}|\,\mathbb{E}_x\mathbb{E}_{y\sim P_\theta(y|x)}\left[-\nabla^2\log P_\theta(y|x)\right] \approx \sum_{x\in\mathcal{D}}\mathbb{E}_{y\sim P_\theta(y|x)}\left[-\nabla^2\log P_\theta(y|x)\right]
\tag{18}
$$

We can further use the identity (Kunstner et al., 2019)

$$
\sum_{x\in\mathcal{D}}\mathbb{E}_{y\sim P_\theta(y|x)}\left[-\nabla^2\log P_\theta(y|x)\right] = \sum_{x\in\mathcal{D}}\mathbb{E}_{y\sim P_\theta(y|x)}\left[\nabla\log P_\theta(y|x)\nabla\log P_\theta(y|x)^\top\right]
\tag{19}
$$

to obtain a formula that does not require us to compute second-order derivatives. This highlights one benefit of the expected Fisher matrix: it is positive-semi-definite by construction.

**Diagonal Approximation** While equation 19 is possible to evaluate in principle, the resulting size of the Hessian matrix is still way too large to store for even small-scale LLMs. We therefore make another approximation $\hat{H}_l$, computing only the diagonal entries in equation 19.

$$
\mathrm{diag}(\hat{H}_l) = \sum_{x\in\mathcal{D}}\mathbb{E}_{y\sim P_\theta(y|x)}\left[(\nabla\log P_\theta(y|x))^2\right]
\tag{20}
$$

The diagonal approximation corresponds to taking the Taylor expansion behind the Laplace approximation for each coordinate separately, which leads to a posterior being a normal distribution with diagonal covariance. However, this distribution is not necessarily the best KL-projection of the Gaussian arising from equation 19 on the space of diagonal Gaussians[9].

---

[8]We use the term expected Fisher to stress the fact that it is different from the empirical Fisher matrix (Kunstner et al., 2019).

[9]Such a projection could be obtained by inverting the formula in equation 19 and taking the diagonal, which is intractable.

**Analytic Solution for Gaussian Aleatoric Noise** If we assume a model where the aleatoric noise is Gaussian with mean zero and fixed variance $\sigma_{\text{obs}}^2$, we can compute the expectation in equation 20 analytically. For the sake of clarity, let us consider the one-dimensional case. The parametric likelihood $P_\theta(y|x)$ is defined as

$$P_\theta(y|x) = \frac{1}{\sqrt{2\pi\sigma_{\text{obs}}^2}} \exp\left(-\frac{1}{2\sigma_{\text{obs}}^2}(y - f_\theta(x))^2\right) .$$

Hence, the gradient of the log-likelihood is:

$$\nabla \log P_\theta(y|x) = \frac{1}{\sigma_{\text{obs}}^2}(y - f_\theta(x))\nabla f_\theta(x) .$$

The expected value of the squared gradient of the log-likelihood is:

$$\mathbb{E}_{y\sim P_\theta(y|x)}\left[(\nabla \log P_\theta(y|x))^2\right] = \frac{(\nabla f_\theta(x))^2}{\sigma_{\text{obs}}^2} \underbrace{\int_{-\infty}^{\infty} \frac{1}{\sigma_{\text{obs}}^2}(y - f_\theta(x))^2 \frac{1}{\sqrt{2\pi\sigma_{\text{obs}}^2}} \exp\left(-\frac{1}{2\sigma_{\text{obs}}^2}(y - f_\theta(x))^2\right) dy}_{I} .$$

$$(21)$$

Now let us focus on the quantity $I$:

$$\begin{aligned}
I =& \frac{1}{\sqrt{2\pi\sigma_{\text{obs}}^2}} \int_{-\infty}^{\infty} (y - f_\theta(x)) \frac{1}{\sigma_{\text{obs}}^2}(y - f_\theta(x)) \exp\left(-\frac{1}{2\sigma_{\text{obs}}^2}(y - f_\theta(x))^2\right) dy \\
=& \frac{1}{\sqrt{2\pi\sigma_{\text{obs}}^2}}\left[-(y - f_\theta(x)) \exp\left(-\frac{1}{2\sigma_{\text{obs}}^2}(y - f_\theta(x))^2\right)\right]_{-\infty}^{\infty} \\
& + \int_{-\infty}^{\infty} \frac{1}{\sqrt{2\pi\sigma_{\text{obs}}^2}} \exp\left(-\frac{1}{2\sigma_{\text{obs}}^2}(y - f_\theta(x))^2\right) dy \\
=& 0 + 1 = 1 .
\end{aligned}$$

$$(22)$$

Hence, we obtained an analytical solution for the quantity of interest:

$$\mathbb{E}_{y\sim P_\theta(y|x)}\left[(\nabla \log P_\theta(y|x))^2\right] = \frac{(\nabla f_\theta(x))^2}{\sigma_{\text{obs}}^2} .$$

$$(23)$$

# D   Additional experimental details

## D.1   Experimental setting

### D.1.1   Computational resources

Our experiments were conducted on one NVIDIA A100 GPU with 80GBs of VRAM. One random run with a GPT2 (124M) bandit, $T = 100$, took about 30 minutes for the fastest bandits (i.e., greedy and dropout) and up to 45 minutes for the bandits that had to compute the Hessian (i.e., Diag. LA and Last LA). We ran 5 different bandit models for 20 random runs during the testing phase, and we repeated those runs for all four tasks. In total, the experiments with GPT2 bandits took around 220 hours of computation. The experiments done with GPT2-XL (1.5B) bandits, $T = 100$, 20 random runs, 5 different bandit models, `hate` task, took around 50 hours of computation.

### D.1.2   Tasks and Data processing

We evaluate the bandit policies on various real-world tasks. These tasks consist of:

- An open-source dataset, called "*measuring hate speech*", which is openly available on *HuggingFace Datasets*[10]. This dataset consists of around 136,000 comments. Each of them is associated with

---

[10]https://huggingface.co/datasets/ucberkeley-dlab/measuring-hate-speech

a continuous score, called the "hate speech score", provided for each comment. Comments with a score $> 0.5$ are considered "toxic" (around 36% of the comments are labeled as toxic, while the others are not toxic). We will refer to this task as `toxic`.

- An open-source dataset, called "*IMDb*" (Maas et al., 2011), which is openly available on *HuggingFace Datasets*[11]. This dataset consists of 50,000 movie reviews. Each of them is associated with a *class*, which can be "positive" or "negative", according to the sentiment expressed in the review. The dataset consists of exactly 50% "positive" reviews and 50% "negative" reviews. We will refer to this task as `imdb`.

- An open-source dataset, called "*Offensive Language Identification*" (Zampieri et al., 2019), which is openly available on *HuggingFace Datasets*[12]. This dataset consists of over 14,000 English tweets. Each of them is classified as "non-offensive" or "offensive". The dataset consists of 66.9% "non-offensive" tweets and 33.1% "offensive" tweets. We will refer to this task as `offensive`.

- An open-source dataset, called "*HatEval*" (Basile et al., 2019), which is openly available on *Hugging-Face Datasets*[13]. This dataset consists of 13,000 English and Spanish tweets, where some of them contain hate speech against immigrants and women. Each of them is classified as "non-hateful" or "hateful". The dataset consists of 58% "non-hateful" tweets and 42% "hateful" tweets. We will refer to this task as `hate`.

For each task, the text is tokenized with the GPT2 Tokenizer. In particular, we rely on the implementation provided by HuggingFace[14]. For each task, we selected a maximum number of tokens, and we applied left padding for the comments with lengths less than the selected number of tokens. For the `imdb` task, we selected 256 as the maximum number of tokens; for the `toxic` task, we selected 128 as the maximum number of tokens; for the `hate` and `offensive` tasks, we selected 64 as the maximum number of tokens. The difference is due to the difference in the length of each data point of those datasets.

These tasks are framed as a contextual bandit problem as follows. The context is the text input, and the actions available to the agent are "not publish" or "publish". For each time step, the agent will observe a batch of $B = 32$ comments. The reward function is the following: if the agent decides not to publish a comment, a reward of 0.5 is observed regardless of the actual toxicity of the comment. If the agent publishes a non-toxic comment, a reward of 1 is observed. If the agent publishes a toxic comment, a reward of -0.5 is observed. This asymmetric reward function represents a possible example suitable for this real-world scenario. The goal of the agent is to minimize the regret (compared to a clairvoyant optimal publishing policy) over time by learning to make optimal publish/not publish decisions based on the text content. For the `imdb` task, we consider "negative" reviews as toxic content; for the `offensive` task, we consider "offensive" tweets as toxic content; for the `hate` task, we consider "hateful" tweets as toxic content.

### D.1.3 Bandit models

To investigate the role of uncertainty in LLM bandits, we compare different TS variants with the greedy baseline. Every bandit model is initialized as in Eq. 1, with a pre-trained GPT2 model (Radford et al., 2019) (either the base one with 124M parameters, or the XL version with 1.5B parameters). We use the implementation provided by the HuggingFace library[15]. GPT2 is a Causal Language Model, which means that the attention of each token can only look to the current token and the previous ones. For our purposes, we remove the final classification head, and we take the embedded features of the last token. This is because the last token can look to all the tokens in the sentence with the attention mechanism. In the GPT2 model, this final feature vector has a dimension of 768. We then add a final linear layer with 2 output neurons, which will be trained to solve our regression task. All the parameters are updated during training. We train each model at the end of each time step (for 50 epochs when we use GPT2, for 5 epochs when we

---

[11]https://huggingface.co/datasets/imdb
[12]https://huggingface.co/datasets/tweet_eval/viewer/offensive
[13]https://huggingface.co/datasets/tweet_eval/viewer/hate
[14]https://huggingface.co/docs/transformers/model_doc/gpt2#transformers.GPT2Tokenizer
[15]https://huggingface.co/docs/transformers/model_doc/gpt2

---

**Algorithm 3** Dropout TS

---

**Require:** Bandit model $f_\theta$, regularization scale $\lambda$.

1: $\theta_p \leftarrow [\theta_{\text{PT}}, 0]$
2: Initialize $\mathcal{D} \leftarrow \emptyset$
3: **for** time $t = 1, \ldots, T$ **do**
4:      Observe context $x_t^1, \ldots, x_t^B$.
5:      **for** $b = 1, \ldots, B$ **do**
6:          Apply dropout to parameters: $\hat{\theta} \leftarrow \text{dropout}(\theta)$.
7:          Select $a_t^b = \arg\max_a f_{\hat{\theta}}(x_t^b, a)$.
8:          Observe reward $r_t^b$.
9:      **end for**
10:     Create $\mathcal{D}_t = \{(x_t^1, a_t^1, r_t^1), \ldots, (x_t^B, a_t^B, r_t^B)\}$
11:     Add to the observed dataset $\mathcal{D} \leftarrow \mathcal{D} \cup \mathcal{D}_t$.
12:     Train the model $f_\theta$ with Adam optimizer for 50 epochs minimizing:

$$\mathcal{L}(\theta^{(t)}; \mathcal{D}_t) = \sum_{b=1}^B (r_t^b - f_{\theta^{(t)}}(x_t^b, a_t^b))^2 + \lambda ||\theta^{(t)} - \theta^{(t-1)}||_2^2$$

13: **end for**

---

use GPT2-XL) with the *Adam* optimizer (Kingma & Ba, 2014), with learning rate set to $3 \cdot 10^{-5}$. As TS variants, we include dropout, Diagonal Fisher LA (which we will call Diag. LA), LA with full Hessian on the last layer (which we will call Last LA), and Epinet TS. In the following paragraphs, we describe in further detail the different TS bandit models.

**Dropout TS** The *dropout* TS differs from standard supervised learning because we still apply dropout during the action selection phase (Gal & Ghahramani, 2016; Riquelme et al., 2018). Using dropout in this phase, we randomly select (according to the dropout probability) a set of parameters $\hat{\theta}$ to use. This procedure can be seen approximately as sampling parameters from a posterior distribution: $\hat{\theta} \sim P(\theta|\mathcal{D})$. With such approximate posterior distribution, we can apply Thompson Sampling. After the action selection phase, we observe the rewards for the selected action, and we update all the parameters of the model by minimizing a MSE loss. A detailed explanation of the Dropout TS is provided in Algorithm 3.

**Diag. LA TS** With the Diag. LA TS, we have to maintain a diagonal Hessian for all the weights. At the beginning, the Hessian is initialized as a diagonal matrix with all the entries equal to the inverse of the prior variance $\sigma_p^2$. Then, action selection is performed by sampling every time a different set of weights. Finally, a maximum-a-posteriori loss is minimized. Once we finished the training phase, we obtain a set of weights which we call $\theta_{\text{MAP}}$. We update our Hessian using the recursive formula and the expected Fisher approximation. With these quantities, we update the posterior distribution $P(\theta|\mathcal{D})$ as $P(\theta|\mathcal{D}) = \mathcal{N}(\theta_{\text{MAP}}, H^{-1})$. Then, we can re-start observing context and selecting actions, repeating the loop. An algorithmic description of this bandit procedure is provided in Algorithm 4.

**Last LA TS** With the Last LA TS, we have to maintain a full Hessian only for the last layer parameters. At the beginning, the Hessian is initialized as a diagonal matrix with all the entries equal to the inverse of the prior variance $\sigma_p^2$ and prior weights equal to zero. Then, action selection is performed by sampling every time a different set of weights for the last layer, while all the other parameters stay fixed. Finally, a regularized MSE loss is minimized. Once we finished the training phase, we obtain a set of weights which we call $\theta_{\text{MAP}}$ for the last layer. We update our Hessian using the recursive formula and computing the full Hessian on the likelihood part of the loss. With these quantities, we update the posterior distribution on the last layer parameters $P(\theta|\mathcal{D})$ as $P(\theta|\mathcal{D}) = \mathcal{N}(\theta_{\text{MAP}}, H^{-1})$. Then, we can re-start observing context and selecting actions, repeating the loop. An algorithmic description of this bandit procedure is provided in Algorithm 5.

---

**Algorithm 4** Diag. LA TS

---

**Require:** Bandit model $f_\theta$, prior variance $\sigma_p^2$, observation variance $\sigma_{\text{obs}}^2$.

1: $\theta_p \leftarrow [\theta_{\text{PT}}, 0]$
2: $H^{(1:0)} = \text{diag}(1/\sigma_p^2)$
3: $P(\theta) = P(\theta|\emptyset) = \mathcal{N}(\theta_p, H^{-1})$
4: Initialize $\mathcal{D} \leftarrow \emptyset$
5: **for** time $t = 1, \ldots, T$ **do**
6:     Observe context $x_t^1, \ldots, x_t^B$.
7:     **for** $b = 1, \ldots, B$ **do**
8:         Sample parameters: $\hat{\theta} \sim P(\theta|\mathcal{D})$.
9:         Select $a_t^b = \arg\max_a f_{\hat{\theta}}(x_t^b, a)$.
10:         Observe reward $r_t^b$.
11:     **end for**
12:     Create $\mathcal{D}_t = \{(x_t^1, a_t^1, r_t^1), \ldots, (x_t^B, a_t^B, r_t^B)\}$
13:     Add to the observed dataset $\mathcal{D} \leftarrow \mathcal{D} \cup \mathcal{D}_t$.
14:     Train the model $f_\theta$ with Adam optimizer for 50 epochs minimizing:

$$\theta_{\text{MAP}} \leftarrow \arg\min_\theta \mathcal{L}(\theta^{(t)}; \mathcal{D}) = \frac{1}{2\sigma_{\text{obs}}^2} \sum_{(x,a,r) \in \mathcal{D}_t} (r - f_{\theta^{(t)}}(x,a))^2 + \frac{1}{2}(\theta^{(t)} - \theta_{\text{MAP}}^{(t-1)})^T H^{(1:t-1)}(\theta^{(t)} - \theta_{\text{MAP}}^{(t-1)})$$

15:     Compute current Hessian of the likelihood: $\text{diag}(\hat{H}_l^{(t)}) = \frac{1}{\sigma_{\text{obs}}^2} \sum_{(x,a) \in \mathcal{D}_t} (\nabla f_\theta(x,a))^2$
16:     Update Hessian $H^{(1:t)} \leftarrow H_l^{(t)} + H^{(1:t-1)}$
17:     Update posterior distribution $P(\theta|\mathcal{D}) \leftarrow \mathcal{N}(\theta_{\text{MAP}}, H^{-1})$, where $H = H^{(1:t)}$
18: **end for**

| Model | Hyperparameter | Range | Selected |
|---|---|---|---|
| Greedy | Regularization factor $\lambda$ | 0.1, 0.5, 1 | 1 |
| Diag. LA TS | Prior variance $\sigma_p^2$ | 0.0001, 0.0005, 0.001, 0.005, 0.01 | 0.0001 |
| | Obs. variance $\sigma_{\text{obs}}^2$ | 0.0001, 0.0005, 0.001, 0.005, 0.01 | 0.01 |
| Last LA TS | Prior variance $\sigma_p^2$ | 0.0001, 0.0005, 0.001, 0.005, 0.01 | 0.01 |
| | Obs. variance $\sigma_{\text{obs}}^2$ | 0.0001, 0.0005, 0.001, 0.005, 0.01 | 0.01 |
| Dropout | Regularization factor $\lambda$ | 0.1, 0.5, 1 | 1 |
| Epinet | Regularization factor $\lambda$ | 0.1, 0.5, 1 | 1 |

Table 1: List of the tuned hyperparameters

**Epinet TS** Regarding the epinet architecture, we follow prior work on epinets (Osband et al., 2023a;b) and select an architecture with a multi-layer perceptron $h$ which is multiplied with a dot product with the epistemic index: $\text{epi}_\eta(x; z) = h_\eta([\text{sg}(\tilde{\pi}_\theta(x)), z])^T z$. In particular, we insert a hidden layer with 256 neurons with GELU activation function (Hendrycks & Gimpel, 2016) (the same as GPT2). The last layer of the epinet is a linear layer with a two-dimensional output of dimension $32 \times 2$. While we conjecture that the choice of the epinet architecture may influence results, exploring epinet design was not the focus of this work. Therefore, we used this architecture, which is inspired by prior work (Osband et al., 2023b). Epinets are a very broad class of neural networks. We believe that our work can shed light on the potential of epinets and we leave further investigation to future work.

### D.2 Hyperparameter tuning

For each GPT2 (124M) bandit model, hyperparameters are tuned on 10 random runs (with different seeds than the testing ones) with $T = 50$ on the `toxic` task. The set of hyperparameters are shown in Table 1. In particular, for all the hyperparameter configurations, we selected the ones that best performed on average, measured by cumulative regret, except for the LA TS models. For the LA TS models, we selected

---

**Algorithm 5** Last LA TS

---

**Require:** Bandit model $f_\theta$, prior variance $\sigma_p^2$, observation variance $\sigma_{\text{obs}}^2$.

1: $\theta_p \leftarrow [\theta_{\text{PT}}, 0]$
2: Initialize the Hessian for the last layer $H^{(1:0)} = \text{diag}(1/\sigma_p^2)$
3: Initialize the prior distribution for the last layer $P(\theta) = P(\theta|\emptyset) = \mathcal{N}(0, H^{-1})$
4: Initialize $\mathcal{D} \leftarrow \emptyset$
5: **for** time $t = 1, \ldots, T$ **do**
6:     Observe context $x_t^1, \ldots, x_t^B$.
7:     **for** $b = 1, \ldots, B$ **do**
8:         Sample last layer parameters: $\hat{\theta} \sim P(\theta|\mathcal{D})$ (the remaining parameters stay fixed).
9:         Select $a_t^b = \arg\max_a f_{\hat{\theta}}(x_t^b, a)$.
10:        Observe reward $r_t^b$.
11:     **end for**
12:     Create $\mathcal{D}_t = \{(x_t^1, a_t^1, r_t^1), \ldots, (x_t^B, a_t^B, r_t^B)\}$
13:     Add to the observed dataset $\mathcal{D} \leftarrow \mathcal{D} \cup \mathcal{D}_t$.
14:     Train the model $f_\theta$ with Adam optimizer for 50 epochs minimizing:

$$\theta_{\text{MAP}} \leftarrow \arg\min_\theta \mathcal{L}(\theta^{(t)}; \mathcal{D}_t) = \sum_{b=1}^{B}(r_t^b - f_{\theta^{(t)}}(x_t^b, a_t^b))^2 + \lambda||\theta^{(t)} - \theta^{(t-1)}||_2^2 \ ,$$

    where $\lambda = \sigma_{\text{obs}}^2/\sigma_p^2$.
15:     Compute the *full* Hessian of the likelihood (only for the last layer parameters):

$$\hat{H}_l^{(t)} = \nabla^2\left(\frac{1}{2\sigma_{\text{obs}}^2}\sum_{(x,a,r)\in\mathcal{D}_t}(r - f_\theta(x,a))^2\right)$$

16:     Update last layer Hessian $H^{(1:t)} \leftarrow H_l^{(t)} + H^{(1:t-1)}$
17:     Update posterior distribution $P(\theta|\mathcal{D}) \leftarrow \mathcal{N}(\theta_{\text{MAP}}, H^{-1})$, where $H = H^{(1:t)}$
18: **end for**

---

the best configuration and the ones that are not statistically different from the best (measured with a t-test, p-value=0.05). Among those configurations, we selected the ones with the highest values for prior and observation variances, in order to induce exploration.

### D.3  Additional ablation experiment on dropout

Among the Thompson Sampling techniques, the Dropout method delivers strong results even without tuning the dropout probability. By relying on the same dropout rate used during pre-training, we are using the same uncertainty that the original model had in learning to generate natural language. Therefore, it is not necessarily the best dropout probability to use in a bandit task. To investigate the importance of this hyperparameter in our experiments, we show dropout TS policies across various dropout probabilities in Figure 8 on the `toxic` bandit task. From these results, the rate originally used for pre-training appears optimal for the bandit task. Also, we notice that, for a smaller value of $p$ ($p = 0.05$), the reduced exploration induces a larger variance, as expected.

### D.4  Additional experiment on model dimension and pre-training

In this experiment, we want to investigate the effect of using a large pre-trained model to initialize the bandit policies in contextual bandits with text as input. To this end, we evaluate different greedy policies on the same four tasks presented in Section 5. To investigate the role of the dimension of the model, we include a smaller transformer with a GPT2-like architecture, with 2 attention blocks, embedding dimension of 128, and 4 heads, for a total of around 7M parameters. The training procedure is analogous to the one of the

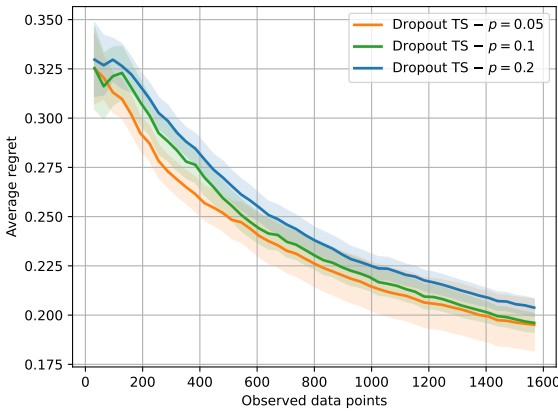

Figure 8: Average regret obtained with different dropout values.

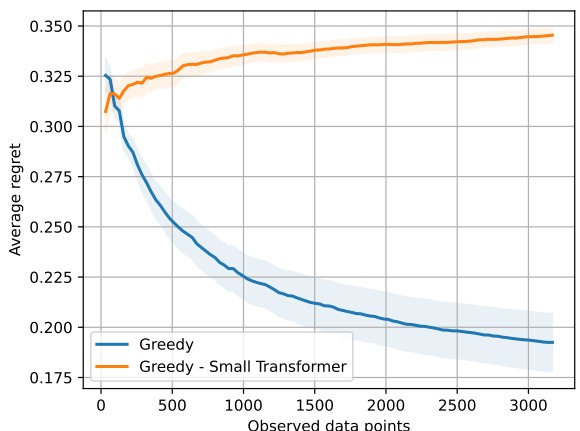

Figure 9: Average regret (± std. err.) obtained on the `toxic` bandit task.

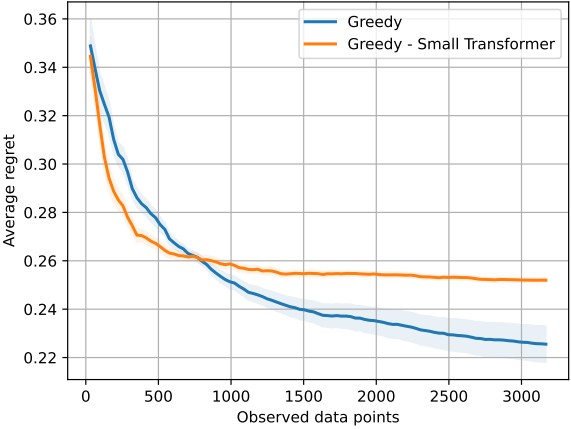

Figure 10: Average regret (± std. err.) obtained on the `imdb` bandit task.

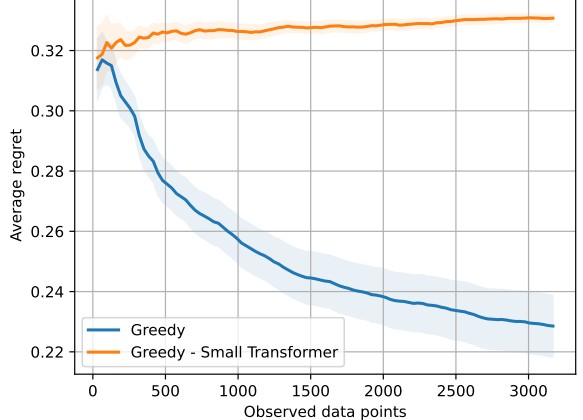

Figure 11: Average regret (± std. err.) obtained on the `offensive` bandit task.

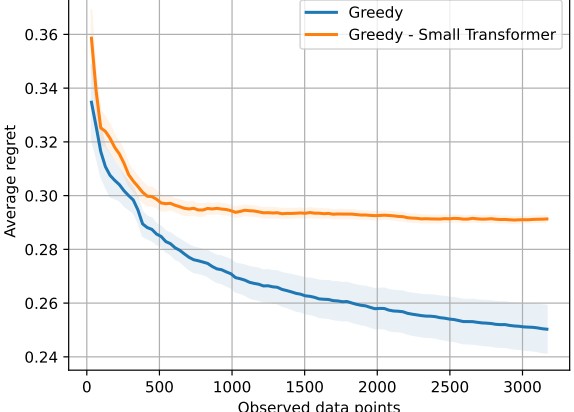

Figure 12: Average regret (± std. err.) obtained on the `hate` bandit task.

---

**Algorithm 6** Epinet TS

---

**Require:** Epinet bandit model $g_{\theta,\eta}$, regularization scale $\lambda$, reference distribution $P_Z$.

1: $\theta_p \leftarrow [\theta_{\mathrm{PT}}, 0]$
2: Initialize $\mathcal{D} \leftarrow \emptyset$
3: **for** time $t = 1, \ldots, T$ **do**
4:     Observe context $x_t^1, \ldots, x_t^B$.
5:     **for** $b = 1, \ldots, B$ **do**
6:         Sample epistemic index $z \sim P_Z$
7:         Select $a_t^b = \arg\max_a g_{\theta,\eta}(x_t^b, a; z)$.
8:         Observe reward $r_t^b$.
9:     **end for**
10:     Create $\mathcal{D}_t = \{(x_t^1, a_t^1, r_t^1), \ldots, (x_t^B, a_t^B, r_t^B)\}$
11:     Add to the observed dataset $\mathcal{D} \leftarrow \mathcal{D} \cup \mathcal{D}_t$.
12:     Train the model $g_{\theta,\eta}$ with Adam optimizer for 50 epochs minimizing:

$$\mathcal{L}(\theta^{(t)}, \eta^{(t)}; \mathcal{D}_t) = \sum_{b=1}^{B} (r_t^b - g_{\theta^{(t)}, \eta^{(t)}}(x_t^b, a_t^b; z^b))^2 + \lambda \|\theta - \theta^{(t-1)}\|_2^2 \, ,$$

    where $z^b \sim_{\mathrm{iid}} P_Z$.
13: **end for**

---

pre-trained greedy policy. The results are shown in Figure 9, 10, 11, 12. These results show how using a pre-trained LLM is essential for solving contextual bandit tasks with text as input. Indeed, the policies that use a small transformer always get stuck with suboptimal actions without learning the real bandit task. By using pre-trained LLMs instead, we can leverage a model that can understand human language (since it was pre-trained on natural language generation) and focus on learning the task-specific reward function using expensive bandit data. This transfer learning approach improves the sample-efficiency of our bandit.

