# OpenReview forum: "On the Importance of Uncertainty in Decision-Making with Large Language Models"
_TMLR — Accepted by TMLR_

### Review · Reviewer_XoFs · 2024-03-30

**Summary Of Contributions:**

The paper investigates the use of large language models (LLMs) as decision making tools, and how equipping LLMs with uncertainty estimates affects their performance. The central claim is that uncertainty estimates play a "fundamental role in bandit tasks with LLMs". This claim is empirically supported by contextual bandit experiments using the "measuring hate speech" dataset with a GPT-2 (124M parameters) LLM with its classification head replaced by a regression head in order to predict rewards. The methods used to equip the LLM with uncertainty estimates are the Laplace approximation, dropout, and epinet. The action selection algorithms tested are the greedy algorithm which does not require uncertainty estimates, and Thompson sampling which requires uncertainty estimates. The experiments show that Thompson sampling outperforms the greedy algorithm across all uncertainty methods.

**Audience:**

No

**Broader Impact Concerns:**

There are no broader impact concerns.

**Claims And Evidence:**

No

**Requested Changes:**

## Critical to recommendation
1. **Run experiments with a large variety of datasets and LLMs**. Studies whose claims are supported purely by empirical evidence need to have comprehensive experiments in order to be convincing.
2. **Either justify or remove non-standard loss function**. Examples of justification include: citing prior work that use it, or run experiments to show that the greedy bandit trained with it is similar to that trained with the standard loss function. I highly recommend removing it and using other methods for scalability. For instance, it is common to freeze the "torso" of the LLM during training and to only update the parameters of the linear regression head, as in Dwaracherla et al. (2024). This effectively turns the LLM torso into a feature extractor, and the extracted features of previous contexts can be cached so that only a small number of LLM forward passes need to be done at each iteration.


## Would strengthen work
1. **Move experiment in appendix to main body and do a similar analysis**.
2. **Downgrade claim.** The word "fundamental" is very strong: the authors should consider downgrading their claim to "uncertainty improves bandit tasks with LLMs".

## Additional thoughts
The authors may notice that the above requested changes only pertain to the question "Are the claims made in the submission supported by accurate, convincing and clear evidence?" The question "Would at least some individuals in TMLR's audience be interested in knowing the findings of this paper?" is subjective, and my answers to that question may not be addressable without changing the research question completely. I advise the authors to consider more interesting problems at the intersection of LLMs and sequential decision making. For example, how does the number of parameters of the LLM affect performance at contextual bandit tasks? Is a 124M parameter model all you need, or does performance keep improving as the models get larger? How are these scaling laws different with Thompson sampling or UCB or the greedy algorithm? Are there contextual bandit tasks in which the generation capabilities of the LLM are useful?

**Strengths And Weaknesses:**

## Strengths:
1. The paper is well organized and well written. The main research question and the approaches to answer it are clear.

## Weaknesses:
I will organize this section in accordance with the TMLR acceptance criteria.

### **Are the claims made in the submission supported by accurate, convincing and clear evidence?**
1. **Empirical evidence is too narrow in coverage to be convincing.** The experiments are run with only 2 datasets (1 relegated to the appendix) and 1 model. The claim that "uncertainty plays a fundamental role in bandit tasks with LLMs" is a large one that is not sufficiently supported by the given experiments. The study needs to include several datasets and several LLMs in order for the results to be plausibly generalizable.

2. **Non-standard loss function.** Equation 3 is a non-standard loss function used to train the greedy bandit, justified by the claim that the standard loss function Equation 2 is not scalable with LLMs. It is not clear whether this loss function is a novel contribution. If so, there is no evidence to believe that this new loss function results in a properly trained greedy bandit (one that is close to that trained by the standard loss function). If this is not a novel contribution and exists in prior continual learning literature, this needs to be clearly cited so that its use is justified.

### **Would some individuals in TMLR's audience be interested in the findings of this paper?**
1. **Expected result.** The study answers the claim: "Equipping a pre-trained language model with uncertainty estimates and using Thompson sampling is better than using the greedy algorithm on 2 language-based contextual bandit tasks." This result is not surprising to the bandits/sequential decision making/probabilistic models audience who would reasonably expect that Thompson sampling is superior to the greedy algorithm no matter the domain.
2. **Uninteresting use of LLMs.** The LLM used here is a GPT-2 (124M parameters) model that is very small by modern standards. Furthermore, the LLM's generation capabilities are not used at all in this paper, as the LLM's classification head is replaced with a regression head (compare this to "Efficient Exploration for LLMs" (Dwaracherla et al., 2024) in which the language generation plays a central role in the decision making loop). For these reasons, the study may not be interesting to the NLP/LLM audience.
3. **Narrow empirical coverage**. Practitioners may not find this study useful in its current form as it only presents results with 1 model and 2 datasets. There is little evidence that the results are generalizable. A practitioner would not be adequately convinced that any particular model and/or uncertainty method is useful for general contextual bandit tasks.

---

> ### Author Response · Authors · 2024-04-30
> **Rebuttal**
>
> We thank the reviewer for the thoughtful review and suggestions. Below we address your comments and requested changes.
>
> # Requested Changes
>
> 1. **Experiments**: We agree that including more experimental evidence would strengthen the paper's claims. However we want to highlight that, as we are focusing on the bandit setting, we believe it is important to run multiple seeds to obtain statistically significant regret results, which limits the size or the number of models we can feasibly experiment with. We believe that good significance in the results arises not only from more datasets, but also from more seeds. While much previous work focus on the former, we focus on experimenting with more seeds because it is a better fit for the bandit setting.
>     Given such constraints, we propose to add experiments with 2 additional datasets and a larger model, 774M or 1.5B pending further assessment of computational cost. We think that the new proposed experimental scope is sufficient. Please do leave a comment if you disagree.
> 2. **Loss function**: We try to clarify this matter in what follows. A standard way of training a model in a supervised setting would be a maximum-a-posteriori (MAP) approach on the whole dataset.
>     At the same time, in a bandit setting, looping on the whole dataset is unfeasible since the dataset is always increasing.
>     However, if we use the loss function in Equation 3, we are doing MAP inference many times, applying Bayes rule anew for each new batch of data. In this way, we perform MAP, which is a well-known established principle in deep learning, while avoiding looping throughout the whole dataset.
>
>     This is the exact same loss used by (Daxberger et al., 2021), described in Appendix C.4.1, in their Continual Learning experiment. Notice that while continual learning and the batch contextual bandit problem have different characteristics (the data shift is exogenous in the former, while it is due to the bandit model that governs the data collection in the latter), the same MAP reasoning applies.
>
>     Daxberger et al., Laplace Redux -- Effortless Bayesian Deep Learning, 2021.
>
> 3. **Move IMDb experiment to the main body**: We thank the reviewer for the valuable advice. We will move the appendix experiment and its analysis to the main text as suggested.
>
> 4. **Downgrade claim**: We agree with the reviewer’s point. We will moderate our claim and say "uncertainty improves bandit tasks with LLMs" rather than "plays a fundamental role".
>
> # Weaknesses
>
> We address the last three weaknesses, since the first two are already addressed by the previous answers.
>
>
> - **Expected result**: While Thompson sampling is known to outperform the greedy approach in classic bandits, scaling it to LLMs is challenging and not yet standard practice. Efficiently estimating uncertainty over LLM parameters and using that to guide exploration requires careful modeling choices. We show how to tractably do this for the first time for LLM bandits.
>     We believe this will be of interest to researchers working on sequential decision-making with LLMs.
>
> - **Use of LLMs**: We respectfully disagree that our use of LLMs is uninteresting. While the LLM we used is not large by current standards, using a smaller model allows us to run multiple seeds per setting, leading to more robust conclusions than many LLM papers. In particular, this aspect is especially vital when it comes to decision-making tasks, where there is inherent stochasticity in the bandit algorithm. Moreover, studying how to use pre-trained LLMs for sequential decision-making is valuable even if generation is not used. This is because we exploit the language-understanding capabilities of LLMs and adapt them to decision-making tasks that take text as input. For these kinds of tasks, it is fundamental to use a model that is pre-trained on a large corpus of text, enabling state-of-the-art language processing capabilities, while it is not necessary to generate text to take actions.
>     We will cite and discuss the interesting recent (Dwaracherla et al. 2024) paper, noting that it addresses a different problem, which is uncertainty-guided reward modeling for alignment.
>
> - **Narrow empirical coverage**: We will add experiments with 2 more datasets and one larger LLM, as mentioned in response to the Requested Change 1, to expand empirical coverage.
>
> # Additional Thoughts
> - We believe LLM bandits are a valuable research direction, both for practical applications (like toxicity detection) and as a stepping stone to more complex settings like text-based RL. We will make this clearer in the paper.
> - Studying scaling laws is an interesting direction for future work but requires computational resources beyond our current capabilities. We will note this in the discussion.
>
> We hope this addresses your main concerns with the paper. We appreciate your suggestions and look forward to strengthening the work by implementing the proposed changes.

---

> > ### Comment · Reviewer_XoFs · 2024-05-03
> > **Response to rebuttal**
> >
> > Thank you for your response.
> >
> > 1. **Experiments**: I acknowledge that 4 datasets, 2 models, 5 uncertainty quantification methods and 20 seeds is a large computational effort, and so I am inclined to say that it is satisfactory.
> >
> > 2. **Loss function**: Your response makes sense. Please include this discussion and citation in the paper.

---

### Review · Reviewer_5dit · 2024-04-05

**Summary Of Contributions:**

The paper proposed to combine various uncertainty based decision making algorithms with a large language model pretrained reward function (or value function). They test out a range of algorithms such as greedy, dropout, laplace approximation and epinet, and benchmark the performance on content moderation task (a contextual bandit task). They show that uncertainty based methods outperform greedy in terms of average regret.

**Audience:**

Yes

**Claims And Evidence:**

Yes

**Requested Changes:**

=== **why pretrained LLMs** ===

Maybe a central question is: why uncertainty based methods with pretrained LLMs? What specific properties of pretrained LLMs are being leveraged in designing the combination?

The paper starts with the premise that it "makes sense" to just combine uncertainty methods with pretrained LLMs. What kind of difference does the fact that the LLMs are "pretrained" make? What if we start with a non-pretrained LLMs, or just a regular neural net, what will the difference be like? Running LLMs does not come without its downside, that is the computational burden. This comparison will illustrate the importance of combining such methods with pretrained LLMs in the first place

=== **the combination does not leverage anything specific about LLM** ===

Throughout the design, there is no specificity to the fact that the underlying models are pretrained LLMs. It is as if the combination with uncertainty based method does not account for the fact that the underlying method is pretrained at all. Maybe a concrete question is, what would you do differently if the underlying model is not pretrained, or not a LLM at all?

=== **narrow evaluation** ===

The eval is only on content moderation task, which is a bit too narrow to illustrate the technical solidity of the approach, ideally there should be more language related benchmarking.

=== **results** ===

It actually comes at little surprise that greedy is outperformed by alternative methods, given evidence in prior work. I think an interesting experiment to run is: if we use non-pretrained LLMs or just a regular transformer based model, how would the performance gap be between different methods? Do we expect pretrained LLMs to enlarge the performance gap or make the gap smaller - there is "argument" that can make the results go either way, and it is interesting to characterize this.

**Strengths And Weaknesses:**

The paper is interesting in that it applies uncertainty based decision making strategies to LLM applications, that of content moderation. This combination feels interesting and might be relevant to application driven audience.

The main weakness of the paper seems to be that the combination seems quite "simplistic" and does not highlight any specific nature of LLMs. There is no technical evidence in the paper that suggests why combining uncertainty based methods with LLM is a sensible choice in the first place, why is better than combining with just regular non-pretrained models. The evaluation is also quite narrow.

So far the results read like "we combine approximate thompson sampling with pretrained LLMs on language bandit tasks and it just works better". This does not shed light on if there is anything special about the fact that the underlying models are pretrained LLMs, nor do we need pretrained models in the first place. Details below.

---

> ### Author Response · Authors · 2024-04-30
> **Rebuttal**
>
> We thank the reviewer for the thoughtful review and feedback on our paper. Below, we address the main concerns and questions.
>
> Regarding the concern that the combination of LLMs and bandits is "simplistic" and does not highlight the specific nature of LLMs:
> - We believe that the combination of LLMs and bandits has significant potential in various applications, such as content moderation, personalized recommendation systems, and dialogue agents. These applications require both the understanding of human language (provided by LLMs) and the ability to make decisions under uncertainty (provided by bandits). Our work is an important step towards enabling such applications.
> - Scaling bandits to work well with LLMs is a non-trivial challenge. For instance, the simplest and best-performing technique to estimate epistemic uncertainty in deep neural networks is the ensemble method. However, in the case of LLMs, this technique is not feasible because it would mean fine-tuning multiple LLMs. Our work addresses this challenge (using alternative epistemic uncertainty estimation techniques) and highlights the effectiveness of the combination (showing a lower regret compared to the greedy algorithm).
>
> Regarding why combining uncertainty-based methods with LLMs is sensible and better than combining with non-pretrained models:
> - Epistemic uncertainty can improve the effectiveness of bandit algorithms. This is because it can enable efficient exploration and a better exploration-exploitation trade-off.
> - Pretraining is necessary for understanding human language. By using pretrained LLMs, we can leverage a model that can understand human language and focus on learning the task-specific reward function using expensive bandit data. This transfer learning approach will improve the sample-efficiency of our bandit.
>
> Regarding the specific properties of pretrained LLMs being leveraged in the combination:
> - Sample efficiency is a key property that we leverage. Unsupervised pretraining data is abundant and cheap, while supervised bandit data is scarce and expensive. By using pretrained LLMs, we can significantly reduce the amount of bandit data needed to learn the bandit reward function. This is because we are focusing on bandit task with text as input. Hence, starting with a model that has no knowledge of natural language would be inefficient.
> - The bandit formalism is a natural way to optimize for sample efficiency, as it explicitly balances exploration and exploitation to minimize regret. By combining LLMs with bandits, we can effectively leverage the sample efficiency of both approaches.
>
> Regarding what we would do differently if the underlying model was not pretrained or not an LLM:
> - Our approach would be similar, with some important differences.
> Let us assume to use a small non-pretrained NN. While in the case of pretrained LLMs we set the prior weights to be the pretrained weights, in the case of a NN initialized from scratch the best option would be to set the prior to be the zero vector. The prior weights encode prior knowledge about the task and pretraining on English text gives us a lot of knowledge about English, which we need for understanding contexts.
> Another difference would be that the data efficiency would be worse without pretraining. Pretraining is crucial for achieving sample efficiency in the context of language-based tasks. Without pretraining, the model would be slower to understand human language, and the bandit algorithms would require more data to learn the reward function effectively.
>
> Regarding the narrow evaluation:
> - We thank the reviewer for the valuable suggestion. We will include 2 more datasets and a larger model in our experimental section.
>
> Regarding the performance gap between different methods when using non-pretrained LLMs or small neural models:
> - We believe that pretraining is crucial for the LLM to understand human language effectively. Without pretraining, all algorithms would likely be inefficient, as they would need to learn to solve the bandit task without having ever seen human text. The performance gap between different methods in this case would be less relevant, as the overall performance would be poorer due to the lack of language understanding.
>
> We hope that these explanations address your concerns and provide a clearer picture of the contributions and significance of our work.
>
> To investigate the need for a large pretrained model, we will include an experiment of a smaller NN initialized from scratch evaluated on one of the bandit tasks of the paper. This will show the improvement that we get by using a pretrained LLM in terms of sample efficiency.

---

### Review · Reviewer_aHHM · 2024-04-22

**Summary Of Contributions:**

The paper proposes doing contextualised bandits on top of LLM features.

**Audience:**

Yes

**Broader Impact Concerns:**

No issues.

**Claims And Evidence:**

Yes

**Requested Changes:**

My main issues are:
i) I'm not sure this is a bandit problem in the first place. The paper needs to include a classification baseline, and demonstrate why it is useful to tackle this problem via bandits.
ii) The presentation needs dramatic improvement in terms of the research goals and setting.
iii) The research claims do need more empirical evidence. Another real-world task should be added. The paper needs to demonstrate why the epistemic/aleatoric stuff is "fundamental".


- In sec 3.0 the connection between LLM and bandits is not concretised sufficiently. I’m not sure what is x (current token or all previous, or something else), a (??), r (??) in this context. I’m not sure how the LLM’s are used, or if they are (Are we still doing token-by-token generation? Since we discarded the softmax, I guess not..? But what are we then doing?) I’m not sure what problem are we solving; what is happening; or why stuff that happens happens. The paper would benefit a lot by having a motivational example to concretise the notation. The paper claims content moderation as example, but does not show how it connects with the methods. One would expect moderation to be a simple binary classification problem, and I don’t see why we need the bandits for that. I don’t understand what is “greedy baseline” for that either. If you want uncertainty, then surely simplest is to do BNNs to predict from the LLM features to the moderation outcome. Can you help me understand?
- In Sec 3.1. it is implied that the LLM weights are also being optimised during the bandit system. Can you comment on the relationship between updating the LLM and updating the bandit reward estimator f: it seems that you are doing two very different things at once, with massive difference in complexity or size. Are you really finetuning the LLM model as you go?
- In sec 3.2. it is implied that you want to do posterior sampling. Again, do you really sample the LLM parameter posterior? This seems unrealistic: can you comment on what kind of estimate are you expecting to get? How can you sample a posterior of billions of parameters?
- In sec 3.2. there is a discussion of exploration and exploitation. I don’t see how this applies to the example of content moderation. How can you “explore” or “exploit” in this setting? What do you “explore”? I think you are just given a piece of text, and you need to approve or disapprove it. This is a simple f>0.5 classification problem and there is nothing to explore or exploit. Surely in bandits you need a settting where you have many actions to take, or you need to take many actions repeatedly to achieve some goal. I also don’t see how the uncertainty plays into this: I don’t think content moderation has much meaningful aleatoric uncertainty. I’m again quite confused of what is the research setting, and what is the research goal, and how do you use LLMs and how do you use bandits. Can you help me understand?
- Sec 4: what is the epistemic uncertainty of the “model”? Uncertainty of what?
- I don’t understand why you can’t do ensembles. Surely you could freeze the LLM and learn an ensemble of single-layer reward predictors on top of this, and train on whatever task data you have. Can you explain what are the bottlenecks assumed to be in this paper?
- What does theta mean in sec 4? Do you apply dropout to both LLM and reward regressor, or only to regressor? The paper says that its both, but I find this very hard to believe. Surely if you have some billion-parameter network and you remove 10% of neurons without the network expecting it, things will get screwed up. Paper also claims it does not add any overhead. Surely you have to now run the dropout many times to get the distribution estimates, which can be very expensive for LLMs.
- Do you really do the Laplace for the LLM parameters?
- Can you overall discuss the finetuning and how expensive it is. Why do you want to finetune the LLM at all? How do you prevent the finetuning from degrading the LLM?
- In experiment I don’t see why the content moderation problem is a bandit problem. It seems that you just have to classify text comments, and that’s it. Where is the bandit angle? In bandits you select from multiple choices iteratively to uncover which choices are good. In the text moderation problem there is no iteration: each comment is decided once and never returned to. Each comment is independent of other comments. There are no multiple choices. It’s not an active learning problem either, since you don’t control what is the next batch. I would then argue this is just a standard single-class classification problem under streaming data.
- What’s the difference between greedy and non-greedy policy. Since this is a single-class problem, the greedy policy just goes for f>0.5 action. So does the other model then choose against the predictor..? This would make no sense. I don’t understand what are you demonstrating in the experiments, or what are you comparing. This is just a classification problem, so how/why does “last LA TS” classify better than “greed”?
- It is implied that greedy policy is doing some kind of exploration in the experiments. What is this exploration? What do you explore? I think it’s just the yes/no action, but why do you need to explore this? Surely your loss will tell you that always deciding “reject” is a bad thing, and would increase the probability of “accept” without any exploration.

**Strengths And Weaknesses:**

S: The overall idea is sensible

W: The presentation is confusing, and I struggled to follow the paper. The research claims, setting and goals could be clarified.

W: The experiments are very limited.

W: The paper seems to just do standard bandit approaches on text tasks. The novelty and significance is limited.

---

> ### Author Response · Authors · 2024-04-30
> **Rebuttal (1/2)**
>
> We thank the reviewer for their valuable feedback and suggestions. Below we address the main concerns raised in the review.
>
> Regarding the main issues:
>
> i) In this paper, we study decision-making with LLMs, and we do that by focusing on the contextual bandit problem with text as context. The motivating example that we use is one of online content moderation, but there are many other use cases where you may want a model to take actions conditioned on natural language. Learning rewards for a bandit problem like that is a regression task, not a classification task. The regression baseline is already included. In the bandit literature, this is called a greedy bandit, meaning we use a vanilla neural network in the supervised learning context to learn the bandit reward, without quantifying our uncertainty about what the reward is.
>
> ii) We will add a motivational example in the paper to better clarify the research goals and setting, following the explanation provided under the point (i).
>
> iii) We thank the reviewer for the suggestion. We will add 2 more datasets and experiments with a larger language model to provide further empirical evidence. Moreover, we will re-phrase our claim: instead of using the word "fundamental", we will state that "uncertainty improves bandit tasks with LLMs".
>
> Regarding the other comments:
> - As described in Sec. 2, the problem we are trying to solve is the contextual bandit problem with text as context. The notation $x,a,r$ is typical in the bandit domain and it is explained in Sec. 2: $x$ is the context (which consists of text), $a$ is the action the bandit needs to take conditioned on $x$, and $r$ is the reward observed after performing action $a$. As stated before, we will add a motivational example in order to clarify the setting. Typical BNN algorithms cannot be applied to LLMs because LLMs have too many weights. Studying ways of measuring epistemic uncertainty that still work with LLMs is one of the main contributions of our paper.

---

> > ### Author Response · Authors · 2024-04-30
> > **Rebuttal (2/2)**
> >
> > - We are dealing with a contextual bandit problem, where the context is natural language. The best models to process natural language are LLMs. Hence, we would like to build an LLM bandit to deal with this problem. We create a reward estimator f as explained in the paper, which has the torso initialized to the LLM and a final regression layer. We finetune the whole LLM bandit as we go to solve the bandit problem. We show that bandits exploiting epistemic uncertainty achieve lower regret.
> > - As stated in the paper (at the end of Sec 3.2): "an exact Bayesian update is computationally infeasible, and we have to resort to approximations". We explain in detail which approximations we use in Sec. 4. A forward pass done on a posterior sample costs the same as a forward pass on a point estimate of the weights.
> > - A contextual bandit with two actions is still a contextual bandit. There is still a lot to learn in bandit with just two actions because there is a huge number of possible contexts, which lead to different outcomes for each action. Exploration-exploitation is needed both because we want to learn this quickly and because greedy contextual bandit algorithms that do not explicitly explore can get stuck, taking sub-optimal actions forever. A classification or regression supervised learning problem would be different because it would be using a fixed dataset as opposed to constructing the dataset by taking actions.
> > - We model the epistemic uncertainty of the model (see Sec. 2), as a probability distribution on the weights $P(\theta|\mathcal{D})$ which we can not compute exactly, but we can estimate with various techniques described in Sec. 4. In other words, the epistemic uncertainty of the model is a probability distribution that reflects our belief about what the weights are.
> > - In this paper we assume that an ensemble (with multiple fine-tuned LLMs) is unfeasible. Creating an ensemble of single-layer reward predictors on top of a single LLM would be feasible, but this is not what is called ensemble in the literature. A very similar model to having more single-layer predictors is the epinet bandit, which we included in our analysis.
> > - $\theta$ is the complete set of weights of the bandit model $f$. We apply dropout to the whole model, and this is feasible for two reasons: 1. the pretrained LLM we are using is pretrained with dropout. This means that the LLM has learned representations that are robust to dropout; 2. Thompson sampling does not require to estimate the distribution, but only to sample from it. Hence, the technique is really simple and with no overhead: we just activate dropout during inference and we select the action suggested by the model with dropout activated. This requires a single forward-pass.
> > -  When we want to use Laplace Approximation for the whole bandit model (which includes an LLM), we must resort to a diagonal approximation (see Sec. 4.2, 4.3. for further detail). However, if we apply Laplace Approx. only on the last layer, we can perform a full Hessian computation.
> > - Finetuning the bandit model on a batch of data costs the same as supervised learning finetuning would cost on the same batch. Finetuning is essential for learning the reward model. Degrading the LLM for next-token prediction does not matter because we are not using the finetuned model to predict the next token, only for predicting the bandit reward.
> > - This is again a question of supervised vs. bandit, which we believe we answered above. However, we want to stress a particular point. The reviewer here suggests that "this is not a bandit problem, it is a supervised problem since there are only two actions". This is not the definition of bandit vs supervised problems. There are bandit problems with 2 actions and supervised problems with 100 classes and vice versa. The real difference is the type of feedback the model observes: in a bandit problem, you only observe the reward of the action you selected. We also definitely do control the next batch.
> > - The difference between greedy and TS is explained in Algorithm 2. In particular, the TS policies perform a sample of the weights of the model from an approximate posterior distribution before taking an action, while the greedy policies use a point estimate of the weights.
> > - Any bandit algorithm, including greedy, has to take some actions, meaning it explores to an extent. However, the exploration properties of the greedy policy are way worse than the Thompson Sampling policy since the greedy policy does not have a notion of epistemic uncertainty.

---

> > > ### Comment · Reviewer_aHHM · 2024-05-05
> > > **resp**
> > >
> > > Thanks for the response.
> > >
> > > In your response you state that the next batch is "controlled". What does this mean? Do you choose which are the next 32 texts somehow? How? I don't see this in any of the algorithm boxes. I also don't see any acquisition functions wrt batching.
> > >
> > > The revised paper has a remark about iterative MAP inference, which seems vague. I think this implies that you use bayesian updating based on the new batch alone, but this needs to be made precise and rigorous, and shown in math notation (appendix is fine). It is easy to make a mistake here, especially if one uses approximative inference.
> > >
> > > I find the revised claims still imprecise:
> > > - "all TS policies consistently outperform greedy..." / "greedy policy is sub-optimal". This does not seem true: in 1.5B case the greedy outperforms TS for a lot of the training regime.
> > > - "we provide comprehensive..". I find this to be not very comprehensive study of contextualised bandits. You only consider one type of bandit method with few different uncertainty estimates.
> > >
> > > With above fixes I think the claims/evidences part is sufficient.
> > >
> > > I still wonder about the "interest to audience" criterion. The paper does show that uncertainty is useful when doing bandits with LLMs, but I don't think it has shown why we would be doing LLM bandits in the first place. Is this useful? The four tasks studied are all examples that could be solved using regular classification methods without sequential RL. Could you argue why this is interesting?

---

> > > > ### Author Response · Authors · 2024-05-08
> > > > **Thanks for the response**
> > > >
> > > > Thanks for the response.
> > > >
> > > > >In your response you state that the next batch is "controlled". What does this mean? Do you choose which are the next 32 texts somehow? How? I don't see this in any of the algorithm boxes. I also don't see any acquisition functions wrt batching.
> > > >
> > > > The actions in the batch are controlled (chosen by the algorithm). The contexts are not controlled. This is the standard contextual bandit setting. Apologies if this was not clear from the paper.
> > > >
> > > > >The revised paper has a remark about iterative MAP inference, which seems vague. I think this implies that you use bayesian updating based on the new batch alone, but this needs to be made precise and rigorous, and shown in math notation (appendix is fine). It is easy to make a mistake here, especially if one uses approximative inference.
> > > >
> > > > We agree with the reviewer. We will add a detailed explanation in math notation in the Appendix.
> > > >
> > > > >I find the revised claims still imprecise:
> > > > "all TS policies consistently outperform greedy..." / "greedy policy is sub-optimal". This does not seem true: in 1.5B case the greedy outperforms TS for a lot of the training regime.
> > > >
> > > > We try to clarify this matter. In the bandit literature, the performance of an algorithm is usually measured with regret. Hence, in this sense, the greedy policy is sub-optimal because it reaches a higher regret at time T. However, the reviewer makes a very interesting point that we will stress more in our analysis: due to under-exploration, the greedy policy initially outperforms the TS ones, but this short-term benefit ends up being harmful in the sense that it prevents the greedy policy from achieving a lower final regret.
> > > >
> > > > We will add this analysis to our paper and mitigate our claims by saying "in terms of final regret" whenever we say "TS outperforms greedy" or "greedy is sub-optimal".
> > > >
> > > > >"we provide comprehensive..". I find this to be not very comprehensive study of contextualised bandits. [...] With above fixes I think the claims/evidences part is sufficient.
> > > >
> > > > Regarding this, we reckon that analyzing 4 different techniques of epistemic uncertainty approximation for LLM bandits is an important and comprehensive effort. At the same time, we acknowledge that this definition may be subjective.
> > > >
> > > > We propose to substitute the word "comprehensive" (which may imply that we tried every possible kind of bandit policies with LLMs) with the word "extensive" (which should imply the important computational effort and extensive research for epistemic uncertainty approximation techniques).
> > > >
> > > > >I still wonder about the "interest to audience" criterion.
> > > >
> > > > We believe that sequential decision-making with text as input is a topic that is very interesting to the community. For this reason, we study contextual bandit with text as input. This has two benefits.
> > > >
> > > > - First, these kinds of contextual bandits have several real-world applications. One possible example we make is the one of the online content moderation scenario, where you only observe the reward for the action you make ("publish" or "not publish") because the reward is given by the users of your platform (hence, in this case, you could not use supervised learning due to partial feedback). Another analogous example could be an online ad placement system. In this hypothetical example, the context is the body (in the form of text) of the ad, and the agent should decide whether to show this news on the webpage or not. According to the action the agent makes, it will observe the corresponding feedback from the users of the platform (e.g., the clickthrough rate). This is another application that is analogous to the example we make (text as context, two actions, partial feedback), which is important in practice, but there may be many others.
> > > > - Second, our work will serve as a stepping stone for studying how to employ epistemic uncertainty for more general decision-making problems (like RL) with text as input. This kind of problem is interesting to the community despite very few recent works actively employing epistemic uncertainty in this domain.
> > > >
> > > > Moreover, TMLR has published several papers on large language models, such as (Singh et al., 2024) and (Yamada et al., 2024), for instance. The journal has also published works on bandits, like (Kang et al., 2024) or (Bian et al., 2024). Given the readership's demonstrated interest in both large language models and bandit algorithms, we believe that the intersection of these two areas, as presented in our paper, will be of significant interest to the TMLR audience.
> > > >
> > > > Singh, et al. "Beyond human data: Scaling self-training for problem-solving with language models.", 2024
> > > >
> > > > Yamada et al., "Evaluating Spatial Understanding of Large Language Models", 2024
> > > >
> > > > Kang et al., "Online Continuous Hyperparameter Optimization for Generalized Linear Contextual Bandits", 2024
> > > >
> > > > Bian et al., "Indexed Minimum Empirical Divergence-Based Algorithms for Linear Bandits", 2024

---

### Author Response · Authors · 2024-05-05
**Revised version**

Dear Reviewers and Action Editor,

We have carefully reviewed and addressed all the comments provided by the reviewers. We have attached our revised paper that addresses the feedback and concerns raised by the reviewers. The main additions and revisions are highlighted in *blue*.

Here we summarize the changes we made:
- Run experiments with a large variety of datasets and LLMs **[reviewers XoFs, 5dit, aHHM]**
  - We added two new tasks and an experiment with a larger language model (1.5B) in our experimental analysis.
- Justification of the loss function **[reviewer XoFs]**
  - We added a discussion and a justification for the loss we are using.
- Downgrade claim **[reviewers XoFs, 5dit, aHHM)**
  - We moderated our claims and now state that "uncertainty improves performance on bandit tasks with LLMs" rather than "plays a fundamental role".
- Add discussion on future work (like scaling laws) **[reviewer XoFs]**
  - We added a discussion on interesting directions for future work.
- Move IMDb experiment to the main body **[reviewer XoFs]**
  - We moved the IMDb experiment to the main body, along with the new experiments we added.
- Cite (Dwaracherla et al., 2024) **[reviewer XoFs]**
  - We cite and discuss the interesting (Dwaracherla et al., 2024) paper in the Prior Work section.
- Experiment with a small transformer **[reviewer 5dit]**
  - We added an ablation study to investigate the role of pre-training and the dimension of the bandit models.
- Add a motivational example to clarify the problem setting **[reviewer aHHM]**
  - We added a motivational example in the introduction, in order to better clarify the problem setting.

---

### Decision · Action_Editor_rnyX · 2024-06-17

**Recommendation:** Accept as is

**Comment:**

This paper combines uncertainty estimation techniques with LLMs bandits methods. The author compare variants of dropout and Laplace approximation, evaluated on decision-making textual tasks.

This is a useful empirical analysis, and all the reviewers agree that this combination of methods is relevant for the TMLR audience. The authors have addressed the main concerns of the reviewers. The last concern that stand out is the lack of apparent novelty, but I think this should not be an important factor in the decision. I thus recommend acceptance.

**Audience:**

Yes

**Claims And Evidence:**

Yes